# Seizures, behavioral deficits, and adverse drug responses in two new genetic mouse models of *HCN1* epileptic encephalopathy

Andrea Merseburg[1,2,3], Jacquelin Kasemir[1,2], Eric W Buss[4], Felix Leroy[4], Tobias Bock[4], Alessandro Porro[5], Anastasia Barnett[4], Simon E Tröder[6], Birgit Engeland[1,2], Malte Stockebrand[1,2], Anna Moroni[5], Steven A Siegelbaum[4], Dirk Isbrandt[1,2,3]*, Bina Santoro[4]*

[1]German Center for Neurodegenerative Diseases (DZNE), Bonn, Germany; [2]University of Cologne, Institute for Molecular and Behavioral Neuroscience, Cologne, Germany; [3]University of Cologne, Center for Molecular Medicine Cologne, Cologne, Germany; [4]Department of Neuroscience, Mortimer B. Zuckerman Mind Brain Behavior Institute, Columbia University, New York, United States; [5]Department of Biosciences, University of Milan, Milan, Italy; [6]In vivo Research Facility, Faculty of Medicine and University Hospital Cologne, University of Cologne, Cologne, Germany

**\*For correspondence:**
dirk.isbrandt@dzne.de (DI);
bs73@columbia.edu (BS)

**Competing interest:** The authors declare that no competing interests exist.

**Abstract** De novo mutations in voltage- and ligand-gated channels have been associated with an increasing number of cases of developmental and epileptic encephalopathies, which often fail to respond to classic antiseizure medications. Here, we examine two knock-in mouse models replicating de novo sequence variations in the human *HCN1* voltage-gated channel gene, p.G391D and p.M153I (*Hcn1*[G380D/+] and *Hcn1*[M142I/+] in mouse), associated with severe drug-resistant neonatal- and childhood-onset epilepsy, respectively. Heterozygous mice from both lines displayed spontaneous generalized tonic–clonic seizures. Animals replicating the p.G391D variant had an overall more severe phenotype, with pronounced alterations in the levels and distribution of HCN1 protein, including disrupted targeting to the axon terminals of basket cell interneurons. In line with clinical reports from patients with pathogenic *HCN1* sequence variations, administration of the antiepileptic Na$^+$ channel antagonists lamotrigine and phenytoin resulted in the paradoxical induction of seizures in both mouse lines, consistent with an impairment in inhibitory neuron function. We also show that these variants can render HCN1 channels unresponsive to classic antagonists, indicating the need to screen mutated channels to identify novel compounds with diverse mechanism of action. Our results underscore the necessity of tailoring effective therapies for specific channel gene variants, and how strongly validated animal models may provide an invaluable tool toward reaching this objective.

## Editor's evaluation

This is an important study into the pathogenic role of two distinct sequence alterations in Hcn1 and extends from insights into ion channel physiology all the way to characterizing the animals' spontaneous seizure phenotype. The authors convincingly show that two clinically relevant sequence alterations in Hcn1 have distinct effects on HCN1 channel trafficking that vary with cellular context. They go on to correlate these changes in trafficking and cellular function to differences in seizure behavior and observe that these changes in Hcn1 lead to paradoxical responses to some antiseizure medications, consistent with clinical observations. Intriguingly, although the corresponding genetic changes

produce a profound epileptic encephalopathy in human patients, mixed effects on cognition are seen in mice. A limitation of the study is that this difference is not investigated in more detail, but the authors have certainly made headway towards understanding the role of Hcn1 in certain genetic epilepsies.

## Introduction

Developmental and epileptic encephalopathies (DEE) are a devastating group of diseases, often with poor response to pharmacological treatment, resulting in a lifelong burden of seizures, developmental delay, and intellectual disability. Since the original discovery over 20 years ago that de novo mutations in voltage-gated ion channels can directly cause early childhood epilepsy syndromes (*Singh et al., 1998*), the number of genes and gene variants associated with DEE has grown exponentially. Current estimates indicate that ~30% of DEE patients carry at least one pathogenic variant in the top 100 known gene candidates for the disease, with about a third of the affected genes falling into the category of voltage- and ligand-gated ion channels (*Noebels, 2017*; *Oyrer et al., 2018*; *Wang and Frankel, 2021*). Despite the wealth of genetic data available and recent efforts to model the effects of the pathogenic variants in genetically modified mice, we understand very little about the mechanisms that underlie the brain and neuronal circuit alterations responsible for seizures or the reasons for their drug resistance.

Here, we focus on seizures and pharmacological response profile associated with mutations in the *HCN1* gene, which encodes a hyperpolarization-activated cyclic nucleotide-regulated non-selective cation channel expressed prominently in the brain (*Santoro and Shah, 2020*). To date, a total of 40 different missense variants in the *HCN1* gene have been reported in patients affected by epilepsy and/or neurodevelopmental disorders (source: HGMD Professional 2020.4). Among these, at least 12 different de novo *HCN1* variants have been linked to DEE (*Nava et al., 2014*; *Butler et al., 2017*; *Marini et al., 2018*; *Wang et al., 2019*).

The functional role of HCN1 channels is tied closely to their distinct subcellular localization in the two classes of neurons where HCN1 protein is primarily expressed, pyramidal neurons in neocortex and hippocampus, and parvalbumin-positive (PV+) interneurons across the brain (*Notomi and Shigemoto, 2004*). Within these neuronal classes, the subcellular localization of HCN1 channel subunits is tightly regulated. In excitatory (pyramidal) neurons, the channel is targeted to the distal portion of the apical dendrites, where it constrains the dendritic integration of excitatory postsynaptic potentials, limiting excitability (*Tsay et al., 2007*; *George et al., 2009*). In inhibitory (PV+) neurons, HCN1 channels are localized exclusively to axonal terminals, where they facilitate rapid action potential (AP) propagation (*Roth and Hu, 2020*) and may regulate GABA release (*Southan et al., 2000*; *Aponte et al., 2006*). Decreased expression or activity of HCN1 channels, as observed in several rodent models of acquired epilepsy, has been thus posited to increase overall network excitability and contribute to epileptogenesis and seizures (*Jung et al., 2011*; *McClelland et al., 2011*; *Brennan et al., 2016*).

In this study, we examine two new genetic mouse models that reproduce de novo HCN1 variants previously shown to be associated with severe forms of neonatal- and childhood-onset epilepsy (*Parrini et al., 2017*; *Marini et al., 2018*; *Fernández-Marmiesse et al., 2019*). A prior attempt at modeling *HCN1*-linked DEE in mice (*Bleakley et al., 2021*) resulted in a relatively mild phenotype, wherein spontaneous seizures could not be systematically documented, despite the presence of epileptiform spikes on electrocorticogram recordings (ECoG). Furthermore, it has been reported that mice with either global or forebrain-restricted deletion of HCN1 also lack spontaneous seizures (*Nolan et al., 2003*; *Nolan et al., 2004*), although seizure susceptibility is increased (*Huang et al., 2009*; *Santoro et al., 2010*). This prompted the question whether there may be some general limitation to modeling *HCN1*-linked DEE in mice. To examine this question and probe further whether and how *HCN1* sequence variations may give rise to DEE, we generated mouse models for two other *HCN1* variants linked to DEE in humans.

We examined variants c.1172G>A/p.Gly391Asp and c.459G>C/p.Met153Ile based on three criteria: severity of disease; occurrence in at least two independent patients with similar epilepsy phenotypes; and differing biophysical effects on channel function when tested in heterologous expression systems (*Marini et al., 2018*). Moreover, clinical reports suggest that seizures are poorly controlled with standard antiseizure medications and that certain anticonvulsants in fact exacerbate seizures in

these patients (*Marini et al., 2018*). Our results show that both de novo variants lead to spontaneous seizures in mice and reproduce the paradoxical pharmacological responses seen in patients, providing new insights into the mechanistic basis for the limitations of current drug therapies.

## Results

Pathogenic sequence variants orthologous to human *HCN1* variants p.G391D and p.M153I were generated using the CRISPR/Cas9 approach in the *Hcn1* gene of inbred C57BL/6J mice (see 'Materials and methods'; *Tröder et al., 2018*). The resulting lines, $Hcn1^{G380D/+}$ and $Hcn1^{M142I/+}$, were maintained using a male heterozygote × female wildtype (WT) breeding scheme and heterozygous animals were compared to WT littermate controls. For simplicity, the equivalent variants in mice and humans will be hereafter cited using the appropriate protein and gene nomenclature formatting, however, omitting positional information numbers (e.g., HCN1-GD or HCN1-MI, and $Hcn1^{GD/+}$ or $Hcn1^{MI/+}$).

### General phenotypic features and gross brain anatomy

No overt differences were noted in pup development, except for consistently smaller body weights at weaning in $Hcn1^{GD/+}$ mice of both sexes (*Figure 1A*). Smaller body weights persisted for the life of $Hcn1^{GD/+}$ animals and were accompanied by an ~12% reduction in adult brain weight with gross brain anatomy otherwise normal (*Figure 1B*). Brain area measurements showed a similar reduction. Thus, both cerebellum and brainstem areas were significantly smaller in $Hcn1^{GD/+}$ mice compared to WT (*Figure 1—figure supplement 1*), with the cerebellum more affected than the brainstem, consistent with the higher expression of HCN1 protein in this region (*Notomi and Shigemoto, 2004*). Reduced mean body weight was also observed in $Hcn1^{MI/+}$ males, but no decrease in brain size was noted in either sex for this line (*Figure 1A and B*).

Survival curves revealed a significant number of premature deaths in both $Hcn1^{GD/+}$ and $Hcn1^{MI/+}$ animals (*Figure 1C*). In many instances, animals were found dead in the cage without having shown prior signs of distress, suggesting sudden death. The survival curves revealed some notable differences in the pattern of death observed between the two lines, both with regards to timing and sex specificity. Most deaths among $Hcn1^{GD/+}$ animals occurred before the age of 3 months (*Figure 1C*, left), with no difference between the sexes (survival rate at 30 weeks: 78% male versus 75% female $Hcn1^{GD/+}$ mice, p=0.369, Fisher's exact probability test). In contrast, for $Hcn1^{MI/+}$ animals, females were more severely affected than males (survival rate at 30 weeks: 86% male versus 55% female $Hcn1^{MI/+}$ mice, p<0.001, Fisher's exact probability test), with most deaths in either sex occurring at an age greater than 3 months (*Figure 1C*, right). These findings suggest important differences in the effects of the two variants on brain function and physiology, perhaps reflective of the divergent effects of the mutations on channel biophysical properties (*Marini et al., 2018*) and HCN1 protein expression (see below).

Given the high expression of HCN1 subunits in cerebellum, both in Purkinje neurons and basket cell axon terminals (*Southan et al., 2000*; *Luján et al., 2005*; *Rinaldi et al., 2013*), as well as the motor phenotype observed in HCN1 global knockout animals (*Nolan et al., 2003*; *Massella et al., 2009*), we quantified basic locomotion in an open-field setting (*Figure 2A*), assessed gait parameters (*Figure 2D*), and tested for motor coordination, learning, and strength using a vertical pole paradigm (*Figure 2G*). $Hcn1^{GD/+}$ and $Hcn1^{MI/+}$ animals showed locomotor hyperactivity in the open field, with significant increases in distance moved and running speed, but a reduction in the center versus border zone occupancy ratio, potentially indicative of increased anxiety-like behavior in both lines (*Figure 2B and C*, *Figure 2—figure supplement 1*). Gait analysis reflected the overall higher running speed of $Hcn1^{GD/+}$ and $Hcn1^{MI/+}$ animals compared to WT littermates (*Figure 2E*). When runs of similar speed were compared, significant changes in the base of support (BOS) were observed in $Hcn1^{GD/+}$ animals, with the front paws' BOS being wider at both speed ranges (*Figure 2F*), and the hind paws' BOS narrower at higher running speeds (*Figure 2—figure supplement 2*). No BOS alterations were noted in $Hcn1^{MI/+}$ animals or any of the other gait parameters measured (*Figure 2F*, *Figure 2—figure supplement 3*). When comparing the ability to climb down a vertical pole, $Hcn1^{GD/+}$ animals displayed significant difficulties, with only 1 of 18 animals able to successfully complete the test during trials 1 and 2, and 2 of 18 animals succeeding in trial 3 (*Figure 2H*, *Figure 2—video 1* and *Figure 2—video 2*). In contrast, $Hcn1^{MI/+}$ animals performed as well as their WT littermates, with all mice successfully

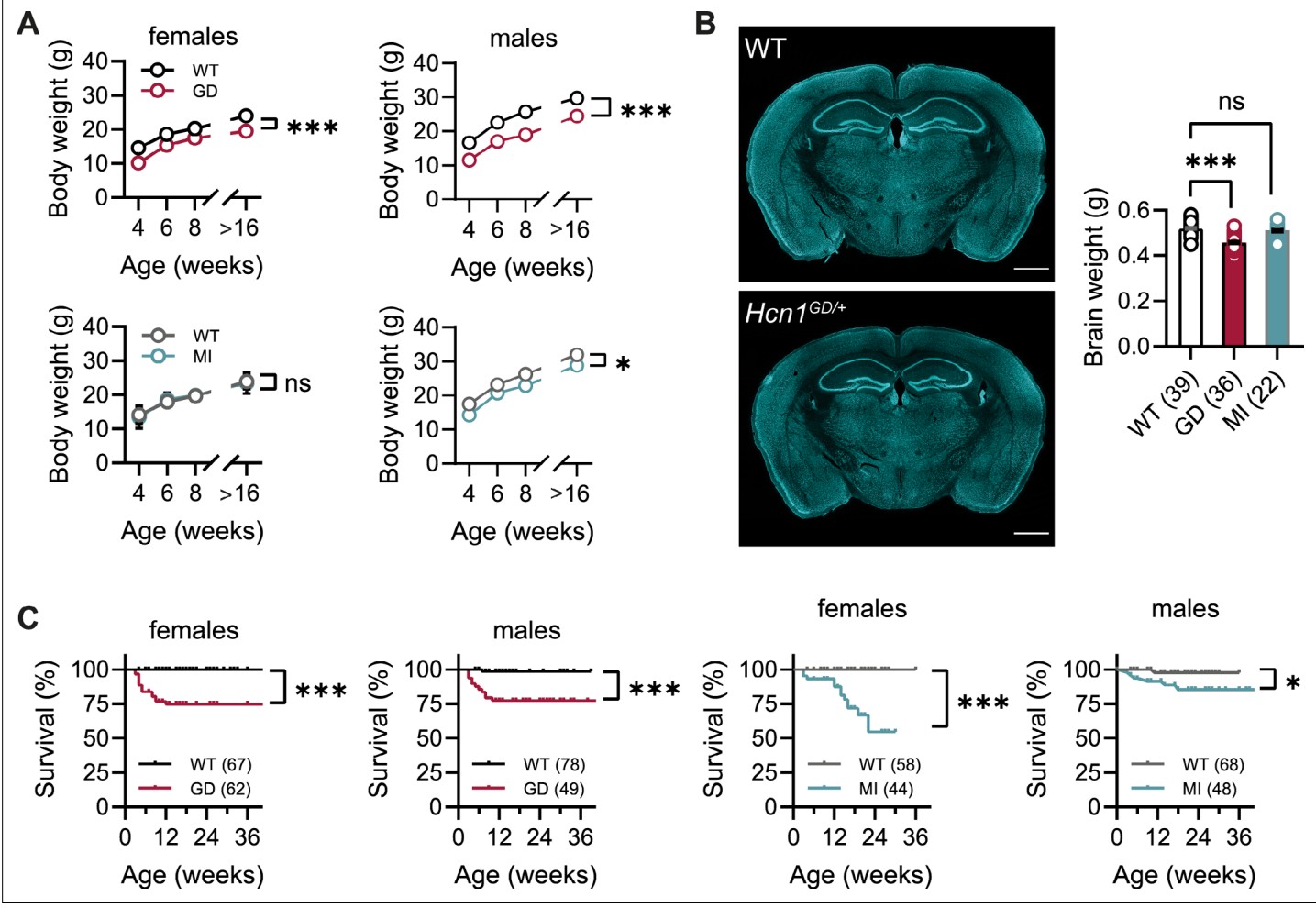

**Figure 1.** Body weight, survival, and general brain morphology in *Hcn1*$^{GD/+}$ and *Hcn1*$^{MI/+}$ mice. (**A**) Reduced body weight post-weaning and through adult life in *Hcn1*$^{GD/+}$ (GD) mice of both sexes, as well as in *Hcn1*$^{MI/+}$ (MI) males (ns = not significant, *p<0.05, ***p<0.001; effect of *genotype*, mixed-effects model having *age × genotype* as grouping factors, GD: WT females n = 11–29, *Hcn1*$^{GD/+}$ females n = 9–23, WT males n = 10–38, *Hcn1*$^{GD/+}$ males n = 9–31; MI: WT females n = 11–28, *Hcn1*$^{MI/+}$ females n = 5–17, WT males n = 5–16, *Hcn1*$^{MI/+}$ males n = 9–20). (**B**) Brain weight in WT, *Hcn1*$^{GD/+}$, and *Hcn1*$^{MI/+}$ mice. Fluorescent Nissl stain of mid-coronal brain sections from WT and *Hcn1*$^{GD/+}$ mice shown on the left (scale bar = 1200 µm). Brain weight shown on the right (WT 0.52 ± 0.01 g; *Hcn1*$^{GD/+}$ 0.46 ± 0.01 g; *Hcn1*$^{MI/+}$ 0.51 ± 0.01 g, ns = not significant; ***p<0.001, after one-way ANOVA and post hoc Holm–Šídák's multiple-comparisons test). Brain weights were measured on PFA-fixed tissue, after removal of both olfactory bulbs. (**C**) Kaplan–Meier survival curves show increased mortality in both *Hcn1*$^{GD/+}$ and *Hcn1*$^{MI/+}$ mice, with a marked sex difference in the case of *Hcn1*$^{MI/+}$ animals. Each dot represents a death event in the colony, scored as 0 if resulting from experimental use and 1 if due to sudden or unprovoked death (survival at 30 weeks, i.e., the last time point where data for all groups were available: *Hcn1*$^{GD/+}$ females = 75%, *Hcn1*$^{GD/+}$ males = 78%, *Hcn1*$^{MI/+}$ females = 55%, *Hcn1*$^{MI/+}$ males = 86%, *p<0.05, ***p<0.001 after log-rank Mantel–Cox test). Data in (**A**) and (**B**) represent mean ± SEM (note that in A smaller error bars may be obscured by the circles representing mean values). Number of animals tested in (**B**) and (**C**) is indicated in parentheses (n).

The online version of this article includes the following figure supplement(s) for figure 1:

**Figure supplement 1.** Cerebellum and brainstem area in *Hcn1*$^{GD/+}$ mice.

completing the task and their average latencies to reach the floor comparable between the two groups (*Figure 2I*, *Figure 2—video 3*, *Figure 2—video 4*).

DEEs are characterized by the presence of significant comorbidities, including developmental delay and cognitive impairments; and patients carrying the *HCN1* p.G391D and p.M153I variants were reported to display severe and mild intellectual disability, respectively (*Marini et al., 2018*). Since HCN1 expression is also very high both in pyramidal neuron dendrites and basket cell terminals throughout the forebrain cortex (*Notomi and Shigemoto, 2004*), we next sought to test the performance of *Hcn1*$^{GD/+}$ and *Hcn1*$^{MI/+}$ mice on behaviors that require forebrain cortical function. Spontaneous alternation is a naturalistic behavior in rodents, which is strongly dependent on intact

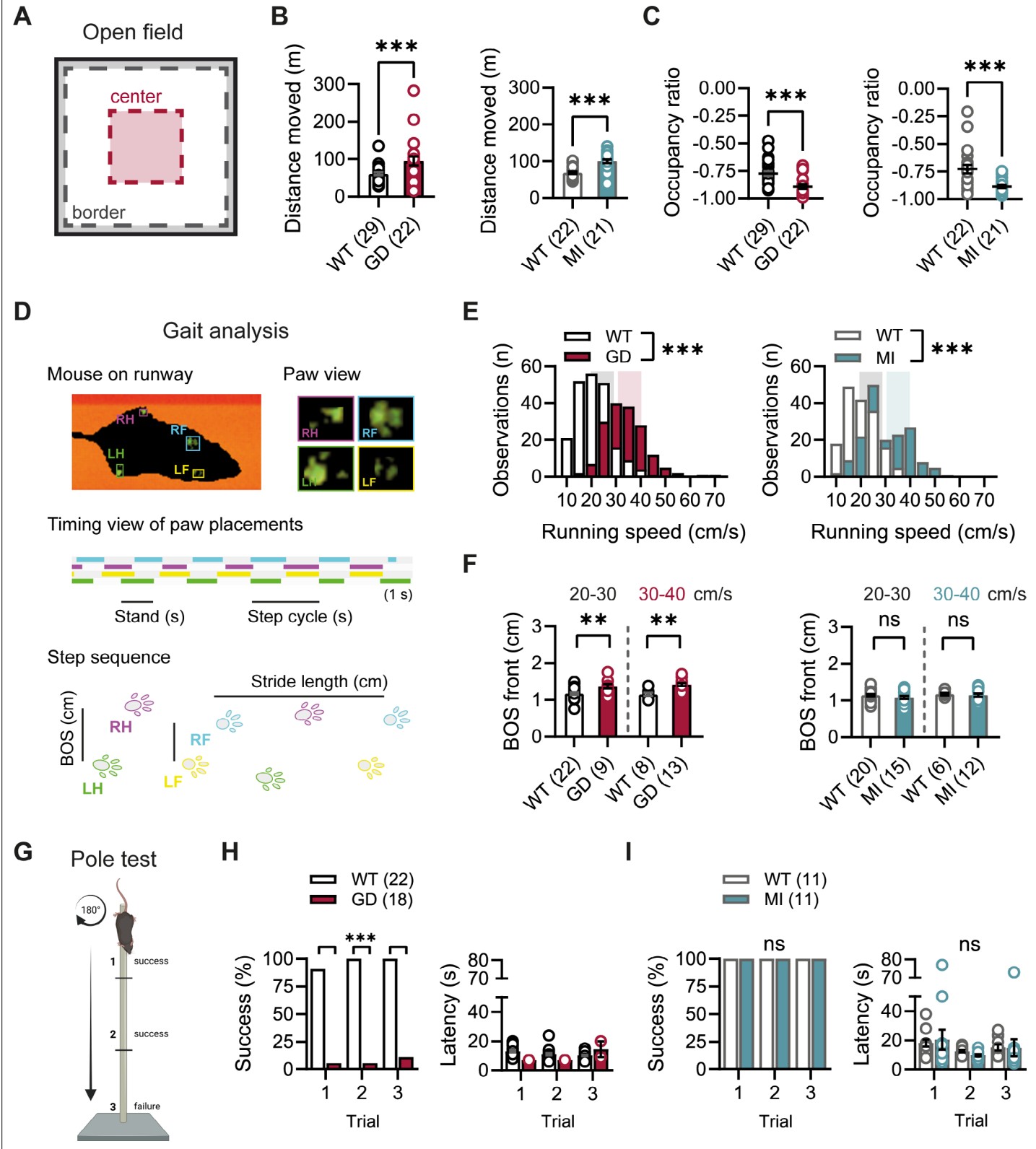

**Figure 2.** Baseline exploratory behavior, gait, and motor function analysis in *Hcn1^GD/+^* and *Hcn1^MI/+^* mice. (**A**) Schematic representation of the open-field arena, with dashed lines depicting the imaginary border (gray) and center zones (red). Occupation ratio calculated as (Center – Border)/(Center + Border), with values <0 representing a preference for the border over the center zone. Mean distance moved was significantly increased (**B**) and mean occupation ratio decreased (**C**) in both *Hcn1^GD/+^* (GD) and *Hcn1^MI/+^* (MI) mice compared to WT littermates (**p<0.01; ***p<0.001, Mann–Whitney *U* test).

*Figure 2 continued on next page*

*Figure 2 continued*

(**D**) Representation of the Catwalk gait analysis system, depicting the mouse on the runway (top left), the paw print view (top right), and below the timing view of consecutive paw placements and step sequence, with the base of support (BOS) representing the distance between the hind and the front paws, respectively. (**E**) Distribution of running speeds of *Hcn1GD/+* or *Hcn1MI/+* mice compared to WT littermates (***p<0.001, Mann–Whitney *U* test). Multiple runs were performed for each animal (GD: WT n = 30 mice, n = 211 runs, mean number of runs per animal = 6.1 ± 0.4; *Hcn1GD/+* n = 21 mice, n = 166 runs, mean number of runs per animal = 7.7 ± 0.4. MI: WT n = 22 mice, n = 166 runs, mean number of runs per animal = 7.6 ± 0.2; *Hcn1MI/+* n = 21 mice, n = 167 runs, mean number of runs per animal = 8.0 ± 0.1). Shaded areas highlight speed range of 20–30 cm/s (gray) and 30–40 cm/s (color). (**F**) BOS of front paws was significantly increased in *Hcn1GD/+* at both speed ranges, but not in *Hcn1MI/+* mice (**p<0.01, ns = not significant, Student's *t*-test). Only runs in the selected speed range were counted and BOS values averaged per animal. Data shown in (**F**) represent mean ± SEM for indicated number of animals. (**G**) Representation of the pole test apparatus (Created with BioRender.com) (**H**) *Hcn1GD/+* mice were severely impaired in motor coordination abilities compared to WT littermates (left; ***p<0.001, Fisher's exact probability test). Latencies to climb down are shown on the right, but due to low numbers of successful *Hcn1GD/+* animals we did not perform statistical analysis. (**I**) All *Hcn1MI/+* mice tested performed successfully in each trial (left; ns = not significant, Fisher's exact probability test) with latencies similar to their WT littermates (right; ns = not significant, two-way repeated-measures ANOVA with *group × trial*). Data in (**B**), (**C**), (**F**), (**H**, right), and (**I**, right) represent mean ± SEM (see *Figure 2—source data 1*, *Figure 2—source data 2*, *Figure 2—source data 3* for numerical values).

The online version of this article includes the following video, source data, and figure supplement(s) for figure 2:

**Source data 1.** Open-field parameters for *Hcn1GD/+* and *Hcn1MI/+* heterozygous mice in comparison to WT littermates.

**Source data 2.** Catwalk gait analysis parameters for *Hcn1GD/+* and *Hcn1MI/+* heterozygous mice in comparison to WT littermates.

**Source data 3.** Vertical pole test parameters for *Hcn1GD/+* and *Hcn1MI/+* heterozygous mice in comparison to WT littermates.

**Figure supplement 1.** Open-field behavior of *Hcn1GD/+* and *Hcn1MI/+* mice.

**Figure supplement 2.** Gait analysis in *Hcn1GD/+* mice.

**Figure supplement 3.** Gait analysis in *Hcn1MI/+* mice.

**Figure 2—video 1.** Pole test recording of a Hcn1GD/+ heterozygous male falling off the pole.

https://elifesciences.org/articles/70826/figures#fig2video1

**Figure 2—video 2.** Pole test recording of a Hcn1GD/+ heterozygous male sliding down the pole.

https://elifesciences.org/articles/70826/figures#fig2video2

**Figure 2—video 3.** Pole test recording of a Hcn1MI/+ heterozygous female.

https://elifesciences.org/articles/70826/figures#fig2video3

**Figure 2—video 4.** Pole test recording of a WT female.

https://elifesciences.org/articles/70826/figures#fig2video4

spatial working memory and involves the activity of several brain regions, including hippocampus and neocortical structures (*Lalonde, 2002*). When tested in a Y maze-based spontaneous alternation task (*Figure 3A*), *Hcn1GD/+* animals failed to display normal alternation behavior, with alternation rates not exceeding the chance level of 50% (*Figure 3B*). In contrast, *Hcn1MI/+* animals showed equal performance compared to WT littermates. Both *Hcn1GD/+* and *Hcn1MI/+* animals needed significantly less time to complete the required 24 arm transitions (*Figure 3C*), consistent with the locomotor hyperactivity observed in these mice. We next tested the mice in an object recognition memory task using a 24 hr interval between training and testing (*Figure 3D*), a paradigm that assesses long-term memory and also requires the activity of multiple cortical regions (*Barker and Warburton, 2011*; *Haettig et al., 2011*). We found that mutant mice from both lines were able to distinguish the novel object from the familiar object, as reflected in positive discrimination indices (D.I.) (*Figure 3E*). *Hcn1GD/+* animals also spent a significantly higher time exploring the novel object, compared to their WT littermates, although the meaning of this finding is unclear.

In summary, our results reveal a complex effect of the two variants on phenotype, with an overall more severe impact on the general fitness of *Hcn1GD/+* compared to *Hcn1MI/+* animals, reflected in smaller body size, reduced brain size, as well as altered gait and locomotion, reduced motor coordination and impairment in certain higher cognitive abilities.

## *Hcn1GD/+* and *Hcn1MI/+* mice show spontaneous convulsive seizures

The sudden death phenotype observed in the two mouse lines, along with the epilepsy syndromes seen in patients carrying *HCN1* p.G391D and p.M153I variants, suggests the animals may be undergoing generalized tonic–clonic seizures (GTCS), which in mice often result in death when escalating into tonic hindlimb extension. One such death event was indeed observed in a female from the

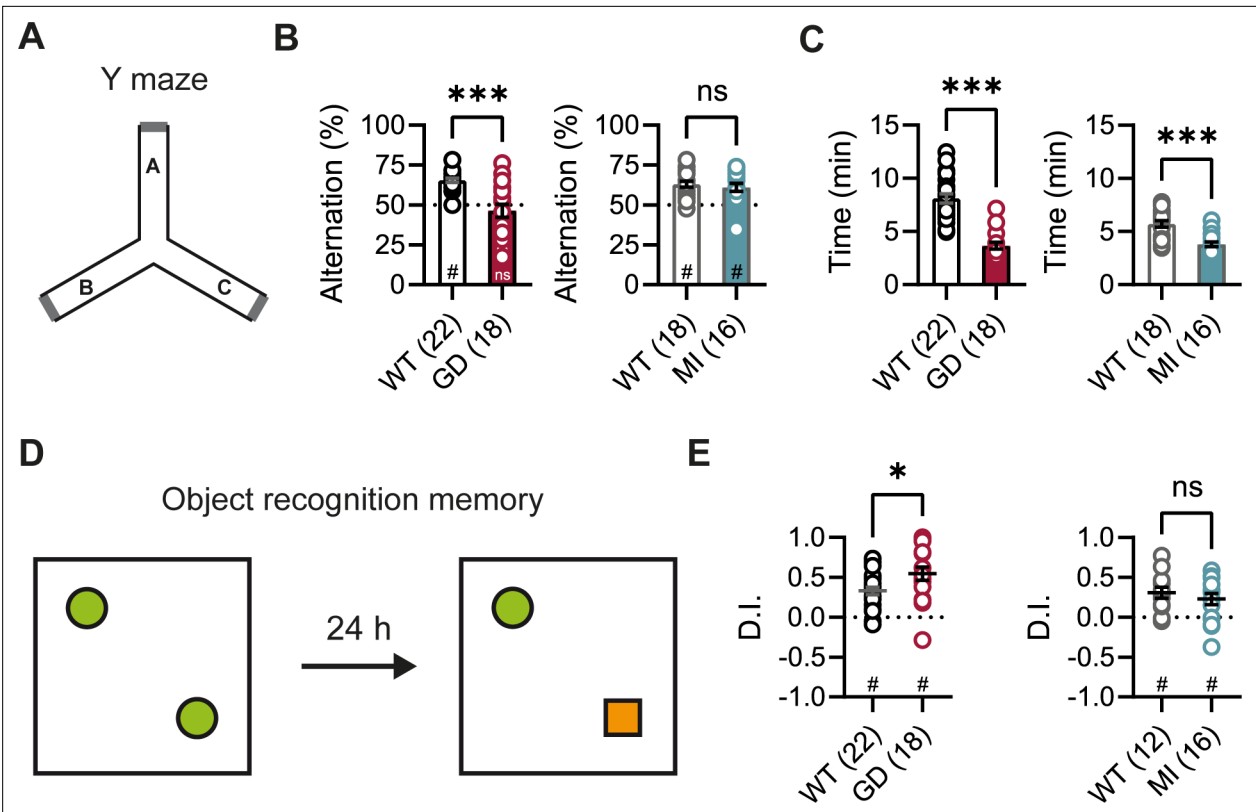

**Figure 3.** Spontaneous alternation and object recognition memory in *Hcn1GD/+* and *Hcn1MI/+* mice. (**A**) Spontaneous alternation test with schematic representation of the Y maze consisting of three equal arms (**A–C**). (**B**) Alternation rates were significantly decreased in *Hcn1GD/+* mice (GD, left), but not in *Hcn1MI/+* mice (MI, right) when compared to their respective WT littermates (***p<0.001, Mann–Whitney *U* test; #p<0.05, one-sample Wilcoxon test to chance level of 50%; ns = not significant). (**C**) Time needed to complete 24 transitions was significantly reduced in both *Hcn1GD/+* and *Hcn1MI/+* mice compared to WT (***p<0.001, Mann–Whitney *U* test). (**D**) Object recognition memory after 24 hr was tested by replacing one familiar object (green circle) with a novel object (orange square). (**E**) Both *Hcn1GD/+* and *Hcn1MI/+* mice showed a preference for the novel object with mean discrimination indices (D.I.) being significantly positive (#p<0.05, one-sample Wilcoxon test to chance level of 0). Mean D.I. was also higher in *Hcn1GD/+*mice compared to WT littermates (*p<0.05, ns = not significant, Mann–Whitney *U* test). Data represent mean ± SEM (see *Figure 3—source data 1*, *Figure 3—source data 2* for numerical values).

The online version of this article includes the following source data for figure 3:

**Source data 1.** Spontaneous alternation in the Y maze.

**Source data 2.** Object recognition memory test.

*Hcn1MI/+* line, and several convulsive seizure events were witnessed in mice from both lines during routine cage inspections. We therefore set out to rigorously quantify the occurrence, frequency, and severity of seizures in *Hcn1GD/+* and *Hcn1MI/+* mice by using chronic video ECoG recordings for periods of 1–2 weeks at a time (*Figure 4*). Adult mice from both lines and sexes were found to display spontaneous GTCSs with distinct electrographic signature on ECoG traces (*Figure 4A and C*) and behavioral manifestation on video recordings (*Figure 4—video 1*, *Figure 4—video 2*).

We recorded from a total of n = 8 *Hcn1GD/+* animals (two males, six females) 56–91 days old to avoid selecting against animals with early death; and a total of n = 12 *Hcn1MI/+* animals (six males, six females) 80–130 days old to capture the period during which sudden death is most frequently seen in females. Notably, despite the higher occurrence of death in females, seizures were observed with similar frequency in males and females from the *Hcn1MI/+* line (*Figure 4D*). Seizure events were comparatively infrequent, ranging from 0.6 to 4.1 seizures per week in *Hcn1GD/+* mice (with 2/8 individuals failing to show seizures during the time recorded) and from 0.6 to 5.1 seizures per week in *Hcn1MI/+* mice (with 2/12 individuals displaying no seizures during the recorded period). Despite the overall more severe phenotype of *Hcn1GD/+* animals, based on body weight, brain size, behavioral performance, and general appearance, when rated on a modified Racine scale (see 'Materials

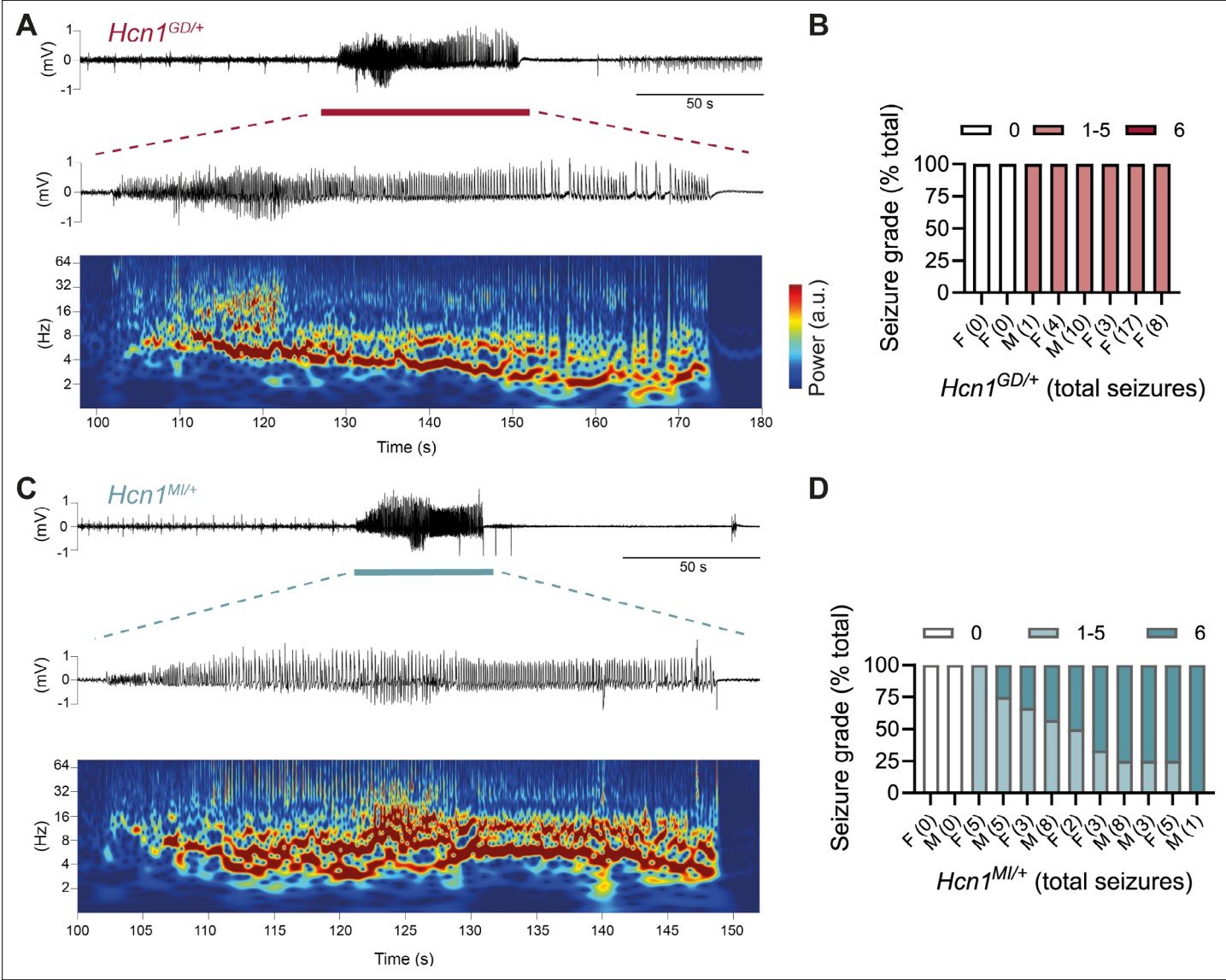

**Figure 4.** Spontaneous convulsive seizures in *Hcn1GD/+* and *Hcn1MI/+* mice. (**A**) Example of a spontaneous seizure recorded in an *Hcn1GD/+* mouse. Upper trace shows electrocorticogram (ECoG) signal of a behaviorally noted grade 4 seizure (red bar), with expanded trace and corresponding time–frequency spectrogram shown below. (**B**) Highest seizure grade reached (plotted as percentage of total seizure number) for all individual *Hcn1GD/+* mice examined, with total number of seizure events observed during the recording period shown in parentheses. 2/8 animals did not exhibit spontaneous seizures during the recording period of 19 ± 3 days. Mean seizure frequency for *Hcn1GD/+* animals was 0.33 ± 0.08 /day. (**C**) Same as in (**A**) showing a grade 6 seizure in an *Hcn1MI/+* mouse. (**D**) Highest seizure grade reached and total seizure event number shown for all individual *Hcn1MI/+* mice. 2/12 animals did not exhibit spontaneous seizures during the recording period of 11 ± 1 days. Of the 10 individuals with seizures, 9 animals reached a seizure grade of 6, indicating an overall higher severity of the spontaneous seizures in *Hcn1MI/+* animals. Mean seizure frequency for *Hcn1MI/+* animals was 0.42 ± 0.07 /day (females, 0.40 ± 0.08 /day; males, 0.45 ± 0.11 /day; p=0.7).

The online version of this article includes the following video and figure supplement(s) for figure 4:

**Figure supplement 1.** Histological markers show hippocampal changes associated with epilepsy in *Hcn1GD/+* and *Hcn1MI/+* mice.

**Figure 4—video 1.** Video - electrocorticogram (ECoG) recording of a spontaneous seizure in Hcn1GD/+ heterozygous female.
https://elifesciences.org/articles/70826/figures#fig4video1

**Figure 4—video 2.** Video -electrocorticogram (ECoG) recording of a spontaneous seizure in Hcn1MI/+ heterozygous male.
https://elifesciences.org/articles/70826/figures#fig4video2

and methods') we found that animals from the *Hcn1*<sup>GD/+</sup> line had on average lower maximum grade seizures compared to *Hcn1*<sup>MI/+</sup> animals (*Figure 4B and D*).

As GTCS in mice usually reflect involvement of the limbic system, including the hippocampus, we sought to additionally characterize the epilepsy phenotype of *Hcn1*<sup>GD/+</sup> and *Hcn1*<sup>MI/+</sup> mice by performing morphological analysis and immunohistochemical labeling for known hippocampal markers of epilepsy. In line with our video ECoG findings, we observed upregulation of neuropeptide Y in dentate gyrus mossy fibers, alterations in granule cell layer morphology and extracellular matrix deposition in the dentate gyrus, as well as increased hippocampal gliosis in a majority of mutant animals from both lines (*Figure 4—figure supplement 1*). Such findings are consistent with the presence of spontaneous seizures and altered excitability within the hippocampal circuit (*Houser, 1990*; *Marksteiner et al., 1990*; *Stringer, 1996*; *Magana-Poveda et al., 2017*). While the site of seizure onset may differ between species, with a more prevalent role of neocortex in human compared to mouse, these results confirm that *Hcn1*<sup>GD/+</sup> and *Hcn1*<sup>MI/+</sup> mice are an overall appropriate model for human *HCN1*-associated DEE, and that the two variants are causal to the epilepsy phenotype.

## Effects of mutations on HCN1 expression and the intrinsic properties of hippocampal CA1 pyramidal neurons

Prior in vitro studies of the human p.G391D and p.M153I variants using heterologous expression systems have demonstrated the functional impact of these mutations on HCN1 channels and the mixed hyperpolarization-activated Na$^+$/K$^+$ current they generate, known as $I_h$ (*Marini et al., 2018*). HCN1 subunits with the G391D mutation fail to express current on their own, but when combined with WT subunits (as occurs in patients and heterozygote *Hcn1*<sup>GD/+</sup> mice) yield, in the ~20% fraction of cells that show measurable currents, a 'leaky' channel with significantly decreased current density but a greatly increased voltage- and time-independent inward, depolarizing current component. In contrast, the M153I mutation only slightly reduces current density, but accelerates the opening kinetics and shifts the voltage dependence of the HCN1 channel to more depolarized potentials (midpoint of activation is shifted by +36 mV for homomeric mutant channels and by +12 mV for heteromeric M153I/WT channels). However, it is not known how these mutations affect HCN1 expression and neuronal physiology in a native environment, which includes both the HCN channel auxiliary subunit TRIP8b (*Lewis et al., 2009*; *Santoro et al., 2009*) and other channel modulators. To address this question, we performed immunohistochemical labeling of HCN1 protein and patch-clamp recordings from CA1 pyramidal neurons in *Hcn1*<sup>GD/+</sup> and *Hcn1*<sup>MI/+</sup> mice and compared their properties with WT littermate controls.

HCN1 antibody labeling of the hippocampus in WT animals showed the typical distribution of HCN1 subunits in CA1 pyramidal neurons (*Figure 5A*), where the channel is present in a gradient of increasing expression along the somatodendritic axis of the apical dendrite (*Lörincz et al., 2002*). This gradient of expression was preserved in brains from both lines, although labeling revealed a substantial decrease in overall HCN1 protein levels in *Hcn1*<sup>GD/+</sup> brains, with a smaller but still significant decrease observed in *Hcn1*<sup>MI/+</sup> brains (*Figure 5B*). In whole-cell patch-clamp recordings, such results were matched by a substantial reduction in the voltage sag observed in response to hyperpolarizing current injections, a hallmark of $I_h$ activity in neurons (*Figure 5C and D*). The voltage sag was absent in *Hcn1*<sup>GD/+</sup> CA1 pyramidal neurons and significantly reduced in *Hcn1*<sup>MI/+</sup>. The lack of voltage sag in *Hcn1*<sup>GD/+</sup> neurons could be due to decreased HCN1 protein expression, a dominant negative effect of HCN1-GD subunits on channel function and/or the increased instantaneous component of HCN channels containing HCN1-GD subunits, which would cause the channel to conduct an inward Na$^+$ 'leak' current with no voltage dependence (*Marini et al., 2018*).

As wildtype HCN1 channels are partially open at the resting membrane potential (RMP) of most cells, including CA1 pyramidal neurons, they exert a well-characterized action to shift the RMP to more positive potentials and decrease the input resistance. Surprisingly, despite the loss of HCN1 protein expression, CA1 pyramidal neurons of *Hcn1*<sup>GD/+</sup> mice displayed a significant *depolarization* in the RMP; in contrast, there was no change in RMP in *Hcn1*<sup>MI/+</sup> neurons (*Figure 5E*). The more positive RMP of *Hcn1*<sup>GD/+</sup> mice was unexpected considering the strong reduction in HCN1 protein expression (*Figure 5B*) and hyperpolarized RMP in mice lacking HCN1 channels (*Nolan et al., 2004*). This may reflect the depolarizing action of the increased voltage-independent 'leak' current seen with certain HCN1 channel mutations, including p.G391D and p.M305L (*Bleakley et al., 2021*), and/or secondary

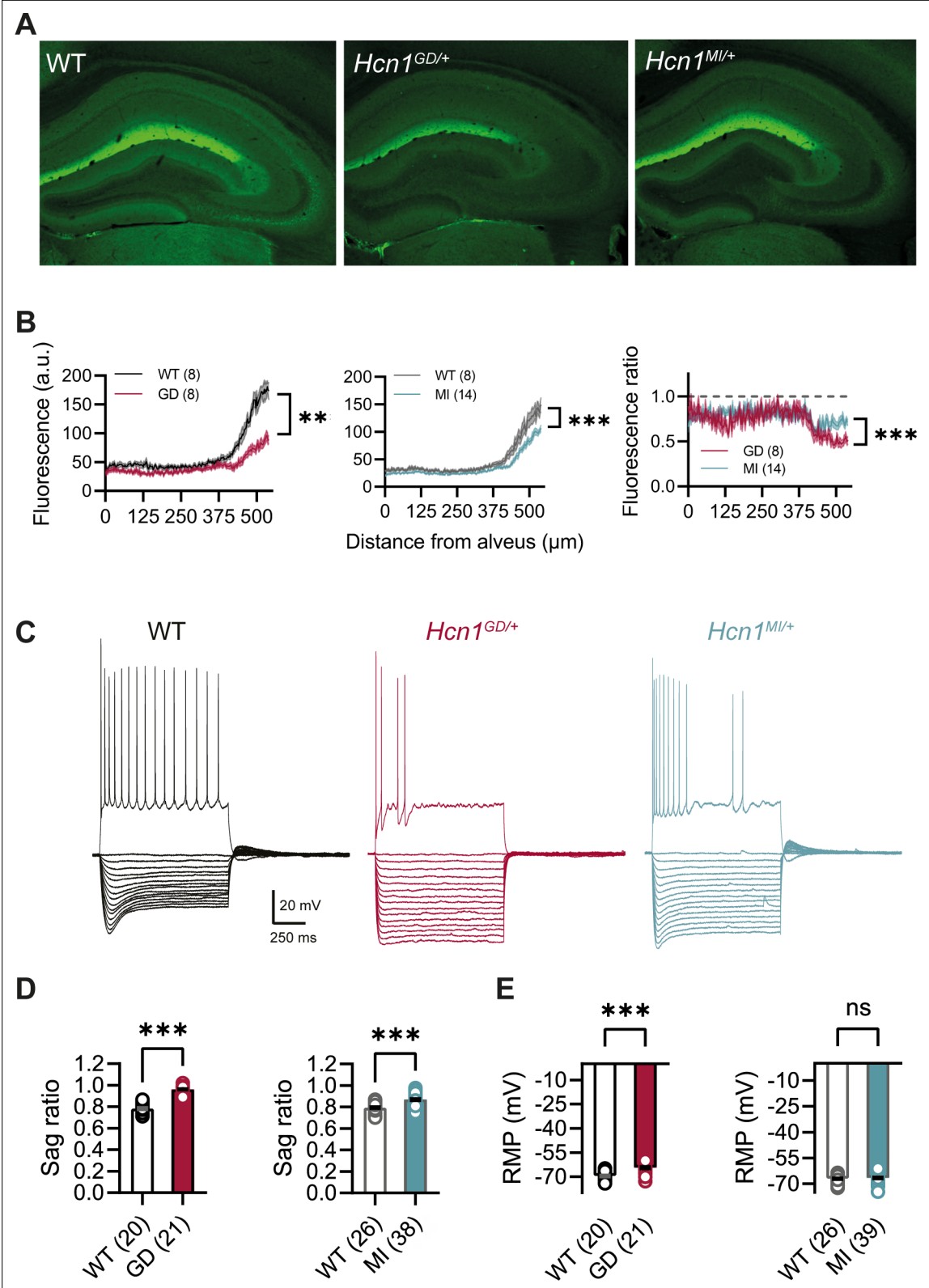

**Figure 5.** HCN1 protein expression and $I_h$-dependent voltage sag in CA1 pyramidal neurons are reduced in $Hcn1^{GD/+}$ and $Hcn1^{MI/+}$ mice. (**A**) Immunofluorescent labeling of HCN1 protein in hippocampus from adult WT, $Hcn1^{GD/+}$, and $Hcn1^{MI/+}$ animals. (**B**) Quantification of fluorescent signal along the somatodendritic axis of pyramidal neurons demonstrating a significant decrease in protein expression in the distal dendrites, the normal site of greatest HCN1 expression (***p<0.001, effect of *genotype × distance* from alveus after two-way repeated-measures ANOVA; number of slices is

*Figure 5 continued on next page*

*Figure 5 continued*

indicated in the graphs; number of animals used: GD WT n = 4, *Hcn1*GD/+ n = 4; MI WT n = 4, *Hcn1*MI/+ n = 7; measurements expressed in arbitrary units [au]). Rightmost panel shows a direct comparison of fluorescence intensity in *Hcn1*GD/+ and *Hcn1*MI/+ animals, measured as a fraction of the corresponding WT littermate control intensities (dashed line at 1); the reduction in HCN1 protein expression was significantly larger in *Hcn1*GD/+ compared to *Hcn1*MI/+ pyramidal neurons (***p<0.001; two-way repeated-measures ANOVA). (**C**) Sample traces from whole-cell current-clamp recordings in hippocampal CA1 pyramidal neurons from WT, *Hcn1*GD/+, and *Hcn1*MI/+ animals. Holding potential was –70 mV for all samples, with a series of hyperpolarizing and depolarizing current steps in 25 pA increments applied in the negative and positive direction (only depolarizing step at +200 pA is shown). (**D**) Voltage sag in response to hyperpolarizing current steps was significantly reduced in both mutant lines (expressed as sag ratio, see 'Materials and methods'; GD: WT = 0.78 ± 0.01 versus *Hcn1*GD/+ = 0.96 ± 0.01; MI: WT = 0.79 ± 0.01 versus *Hcn1*MI/+ = 0.87 ± 0.01; ***p<0.001, Student's *t*-test). (**E**) Resting membrane potential (RMP) was shifted to significantly more positive values in neurons from *Hcn1*GD/+ but not *Hcn1*MI/+ animals, compared to WT littermates (GD: WT = –69.55 ± 0.55 mV versus *Hcn1*GD/+ = –64.43 ± 0.71 mV; MI: WT = –67.08 ± 0.50 mV versus *Hcn1*MI/+ = –66.59 ± 0.51 mV, ***p<0.001, Student's *t*-test; (**D, E**) number of cells is indicated in the bar graphs; number of animals used: GD WT n = 6, *Hcn1*GD/+ n = 5; MI WT n = 6, *Hcn1*MI/+ n = 5). Data represent mean ± SEM.

The online version of this article includes the following source data and figure supplement(s) for figure 5:

**Source data 1.** Action potential (AP) properties of *Hcn1*GD/+ and *Hcn1*MI/+ pyramidal neurons compared to respective WT littermate controls.

**Figure supplement 1.** Impaired action potential (AP) firing in *Hcn1*GD/+ and *Hcn1*MI/+ neurons.

changes in the function of other neuronal conductances. Input resistance was unaltered in both lines (GD: WT = 137.8 ± 7.1 MΩ, n = 20 cells, versus *Hcn1*GD/+ = 153.7 ± 7.8 MΩ, n = 21; MI: WT = 128.4 ± 9.5 MΩ, n = 25, versus *Hcn1*MI/+ = 125.5 ± 7.9 MΩ, n = 38). As input resistance in CA1 pyramidal neurons is typically increased in mice lacking HCN1 channels (*Nolan et al., 2003*; *Nolan et al., 2004*), in the case of *Hcn1*GD/+ mice this may again reflect the offsetting actions of increased 'leak' current and decreased channel expression, or alternatively, offsetting effects due to alterations in the expression of other types of channels. It should be furthermore noted that recordings at the soma might not reveal changes in the input resistance occurring in distal dendrites, where HCN1 expression is highest (*Figure 5A and B*).

The possibility that the expression of mutant HCN1 channels may cause secondary changes in CA1 pyramidal neuron function is consistent with our finding that CA1 cells in both mouse lines show an impaired ability to fire APs, a phenotype not observed with acute application of the HCN channel-specific blocker ZD7288 (*Gasparini and DiFrancesco, 1997*; *Figure 5—figure supplement 1*). Thus, application of positive current steps from a common holding potential of –70 mV revealed an abnormal firing pattern in CA1 neurons from both lines (*Figure 5C*), wherein a majority of *Hcn1*GD/+ neurons and a significant fraction of *Hcn1*MI/+ neurons showed a reduction in the number of APs fired during 1-s-long depolarizing current steps (fraction of neurons with abnormal firing pattern: 16/21 cells = 76% *Hcn1*GD/+ versus 13/35 cells = 37% *Hcn1*MI/+, p=0.006, Fisher's exact probability test showing a larger effect in the former cohort; 1/44 cells = 2% in the combined set of WT littermate neurons). In contrast, rheobase was unaltered in both lines when measured from a holding potential of –70 mV (GD: WT = 100 ± 9.6 pA, n = 20, versus *Hcn1*GD/+ = 114.3 ± 8.0 pA, n = 21; MI: WT = 101.0 ± 6.8 pA, n = 25 versus *Hcn1*MI/+ = 98.6 ± 6.4 pA, n = 38), and no changes were observed in AP threshold, AP peak or AP width (*Figure 5—figure supplement 1*). These findings suggest a lack of secondary changes in voltage-gated Na+ channel function, at least in pyramidal neurons, unlike what has been reported in the case of the p.M305L variant (*Bleakley et al., 2021*). However, alterations in the activity of other channel types (or alterations of Na+ channels in other neuronal classes) remain certainly possible. A microarray-based gene expression screening from WT, *Hcn1*GD/+, and *Hcn1*MI/+ total hippocampal lysates also did not reveal any changes in Na+ channel transcript levels, similar to those observed in *Bleakley et al., 2021* (*Source data 1*; the only candidate with a p-value<0.001 being a K+ channel), although alterations in protein levels or post-translational modulation cannot be ruled out by this approach. Thus, further investigations will be needed to determine how altered conductance in Na+, K+, or other ion channels may potentially contribute to modified neuronal activity in *Hcn1*GD/+ and *Hcn1*MI/+ animals. Finally, no differences were observed between males and females for any of the measures assessed in either mouse line; hence, all reported data were pooled across sexes.

Overall, the observed changes reflect subtle alterations in the intrinsic properties of pyramidal neurons, which do not immediately explain the hyperexcitability and epilepsy phenotype observed in either mouse line. The reduced ability of *Hcn1*GD/+ pyramidal neurons to maintain sustained firing may, in fact, act to limit circuit hyperexcitability in these animals – as reflected in the lower average

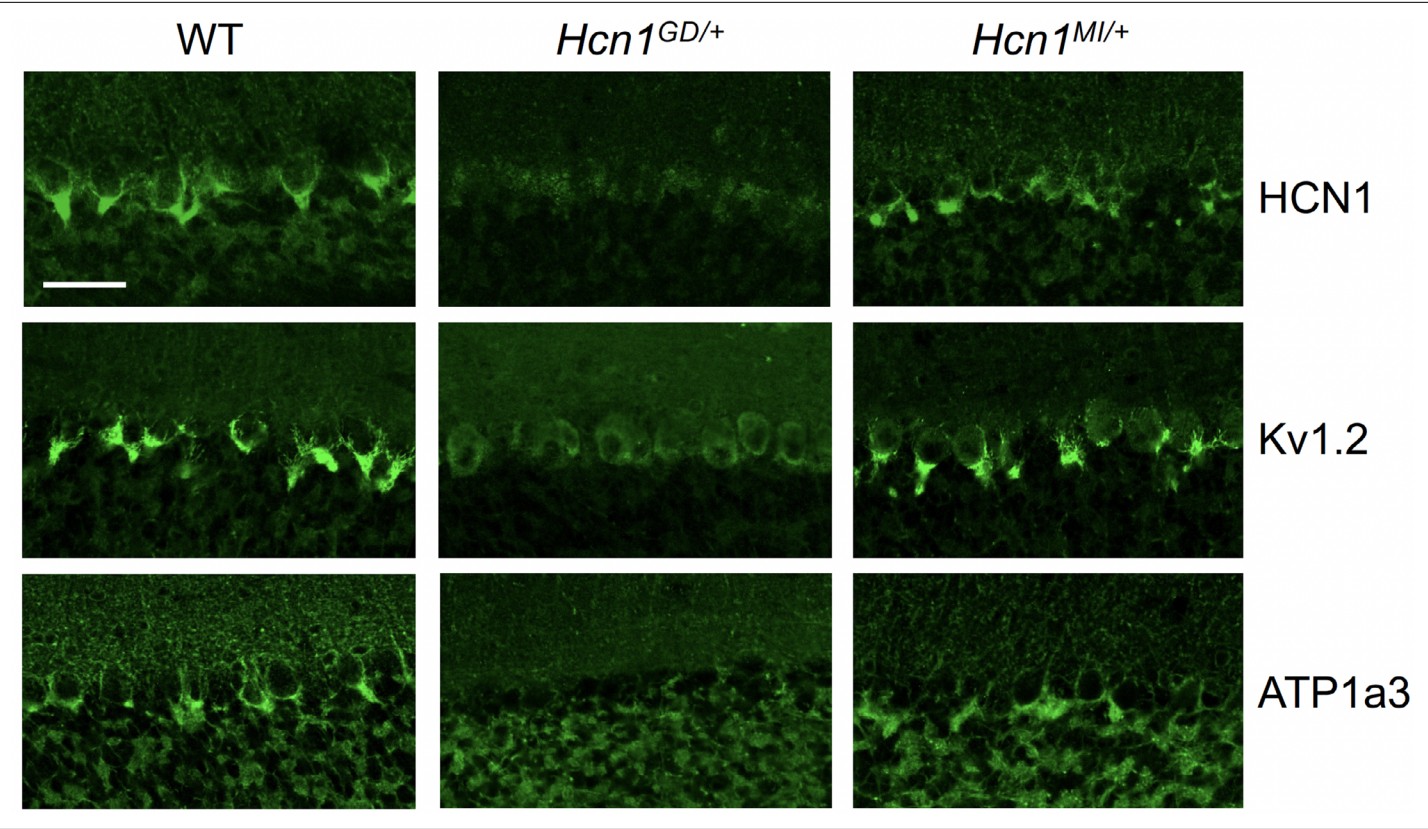

**Figure 6.** Alterations in HCN1 localization in cerebellar basket cell pinceau with accompanying loss of K$_V$1.2 and ATP1a3 in *Hcn1*^GD/+^ but not *Hcn1*^MI/+^ mice. Immunofluorescent labeling of coronal hindbrain sections from adult mouse brain. Detail of the cerebellar cortex of WT, *Hcn1*^GD/+^, and *Hcn1*^MI/+^ mice is shown, displaying the Purkinje cell layer. Top row: HCN1 protein labeling; middle row: K$_V$1.2 protein labeling; bottom row: Na$^+$/K$^+$ pump labeling (protein isoform ATP1a3). The parvalbumin-positive (PV+) interneuron (basket cell) terminal pinceau surrounding the Purkinje cell axon initial segment is labeled by HCN1, ATP1a3, and K$_V$1.2 antibodies in WT and *Hcn1*^MI/+^, while no labeling is present in *Hcn1*^GD/+^ brains. Scale bar = 50 μm in all panels.

The online version of this article includes the following figure supplement(s) for figure 6:

**Figure supplement 1.** K$_V$1.2 protein expression is unaltered in cerebellar pinceau of global HCN1 knockout animals.

maximum seizure grade reached by *Hcn1*^GD/+^ animals both during spontaneous and drug-induced seizure events (***Figure 4B and D***; and see Figure 8C), compared to *Hcn1*^MI/+^ animals. These results suggest that altered activity in non-pyramidal neurons may also play a role in the epilepsy phenotype.

## Severe impairment in the axonal localization of HCN1 protein in *Hcn1*^GD/+^ PV+ interneurons

As HCN1 subunits are highly expressed in PV+ interneurons in the brain, in addition to their expression in pyramidal neurons, we examined whether this expression is altered by the seizure-causing mutations. Whereas the highest site of expression of HCN1 subunits is at the distal portion of apical dendrites of pyramidal neurons, as detailed above, HCN1 is most strongly enriched in the axonal terminals of PV+ interneurons. The exquisite targeting of HCN1 protein to axonal terminals in PV+ interneurons can be particularly appreciated in basket cells from the cerebellar cortex (***Figure 6***). Here, HCN1 protein accumulates in a specialized structure, named the 'pinceau,' representing the axonal termination of basket cells onto the axon initial segment of Purkinje neurons. Surprisingly, at this location, we found a complete loss of HCN1 protein staining in *Hcn1*^GD/+^ animals, while the distribution in *Hcn1*^MI/+^ appeared relatively normal (***Figure 6***, top row). This result is unexpected as levels of HCN1 protein expressed from the WT allele should be normal, providing at least a half dosage in *Hcn1*^GD/+^ cells, and suggests that the mutation may exert a dominant negative effect by co-assembling with WT HCN1 subunits, resulting in improper trafficking and/or premature degradation.

Accumulation of misfolded protein in the endoplasmic reticulum or Golgi apparatus may also impair the trafficking of proteins other than HCN1. As several other proteins are known to specifically localize to the cerebellar pinceau (*Kole et al., 2015*), we used labeling for $K_V1.2$ to assess whether the expression of other channels normally targeted to the pinceau may also be disrupted in *Hcn1^{GD/+}* brains. As shown in *Figure 6* (middle row), $K_V1.2$ labeling of the pinceau was completely lost, similar to HCN1 protein labeling, in *Hcn1^{GD/+}* but not *Hcn1^{MI/+}* brains. The downregulation of $K_V1.2$ does not appear to be due to co-regulation with HCN1, as normal $K_V1.2$ labeling is still present in the cerebellum of HCN1 knockout mice (*Figure 6—figure supplement 1*) and the expression of the two channels at the cerebellar pinceau was shown to be independently regulated (*Kole et al., 2015*). As HCN1 channels in PV+ axonal terminals are thought to functionally interact with the $Na^+/K^+$ pump (*Roth and Hu, 2020*), we further sought to test for the presence of ATP1a3 expression at the pinceau. As shown in *Figure 6* (bottom row), similar to HCN1 and $K_V1.2$, there was a complete loss of ATP1a3 labeling in *Hcn1^{GD/+}* but not *Hcn1^{MI/+}* animals compared to WT controls.

HCN1 channels are expressed in basket cell axons not just in the cerebellum but also throughout the forebrain cortex. Therefore, we next assessed protein labeling in hippocampus, particularly as this region appears to be involved in the mice epileptic phenotype (*Figure 4—figure supplement 1*). Similar to what was observed in the apical dendrites of CA1 pyramidal neurons (*Figure 5A and B*), immunolabeling of HCN1 protein in the axon terminals of hippocampal PV+ interneurons was strongly reduced in *Hcn1^{GD/+}* animals, although not completely absent (*Figure 7A*). Since endogenous protein labeling is unable to distinguish between WT and mutant subunits, we sought to further probe for the trafficking of mutant HCN1 subunits in hippocampal PV+ interneurons using a viral transduction approach. To this end, we used mice that express Cre recombinase specifically in PV+ interneurons (*Pvalb-Cre* line) and stereotaxically injected into the hippocampal CA1 area an adeno-associated virus (AAV) driving Cre-dependent expression of hemagglutinin (HA)-tagged mouse HCN1 subunits (HA-HCN1). As shown in *Figure 7B*, following injection of either HA-HCN1-WT or HA-HCN1-MI, staining with anti-HA antibodies revealed efficient targeting and expression of virus-driven protein in the axonal terminals of PV+ interneurons. In stark contrast, injection of HA-HCN1-GD did not result in detectable levels of HA-tagged protein in axonal terminals, while protein accumulation was observed in the soma of targeted neurons (*Figure 7B*, top row).

Quantitative analysis of HA-tagged protein expression within the pyramidal layer of virus-injected areas confirmed that the virtual absence of axonal staining for HA-HCN1-GD was consistent across multiple animals and injection sites, while HA-HCN1-WT and HA-HCN1-MI expression was comparable (all measurements performed excluding cell somas, see *Figure 7—figure supplement 1* and 'Materials and methods' for details). Thus, the level of signal in sections from HA-WT-injected mice (63 ± 3 arbitrary units [au], n = 16 slices from n = 4 injection sites) was much greater than that observed in sections from HA-GD mice (10 ± 3 au, n = 12 slices from n = injection sites) and similar to levels in sections from HA-MI mice (64 ± 8 au, n = 18 slices from n = 4 injection sites). The signal in HA-GD mice differed significantly from that in HA-WT mice, whereas the signal in HA-MI mice did not differ significantly from that in HA-WT mice (one-way ANOVA with post hoc Holm–Šídák's multiple-comparisons test: effect of *Group*, p<0.001, HA-WT versus HA-GD mice, p<0.001; HA-WT versus HA-MI mice, p=0.837). Co-staining with anti-parvalbumin antibodies furthermore revealed a striking decrease of axonal parvalbumin protein labeling in the targeted CA1 area from brains injected with virus expressing HA-HCN1-GD but not with virus expressing HA-HCN1-WT or HA-HCN1-MI (*Figure 7B*, middle row, and *Figure 7—figure supplements 1 and 2*). Quantitative analysis of parvalbumin protein expression within virus-injected areas (measured excluding cell somas, as described above) again demonstrated similar values in HA-WT mice (75 ± 3 au, n = 6 slices from n = 2 injection sites) and HA-MI mice (83 ± 7 au, n = 6 slices from n = 2 injection sites), but significantly reduced signal intensity in HA-GD-injected animals (48 ± 3 au, n = 12 slices from n = 4 injection sites; one-way ANOVA with post hoc Holm–Šídák's multiple-comparisons test: effect of *Group*, p<0.001, HA-WT versus HA-GD mice, p=0.003; HA-WT versus HA-MI mice, p=0.418).

While the observed loss of parvalbumin, $K_V1.2$ or ATP1a3 labeling in axonal terminals does not distinguish between loss of protein expression and loss of axonal structure or morphology per se, these results underscore the markedly adverse effect of the HCN1-GD variant on PV+ interneuron cellular biology. Overall, our combined findings on native protein expression in the cerebellar pinceau and virus-transduced protein expression in the axonal terminals of hippocampal basket cells suggest

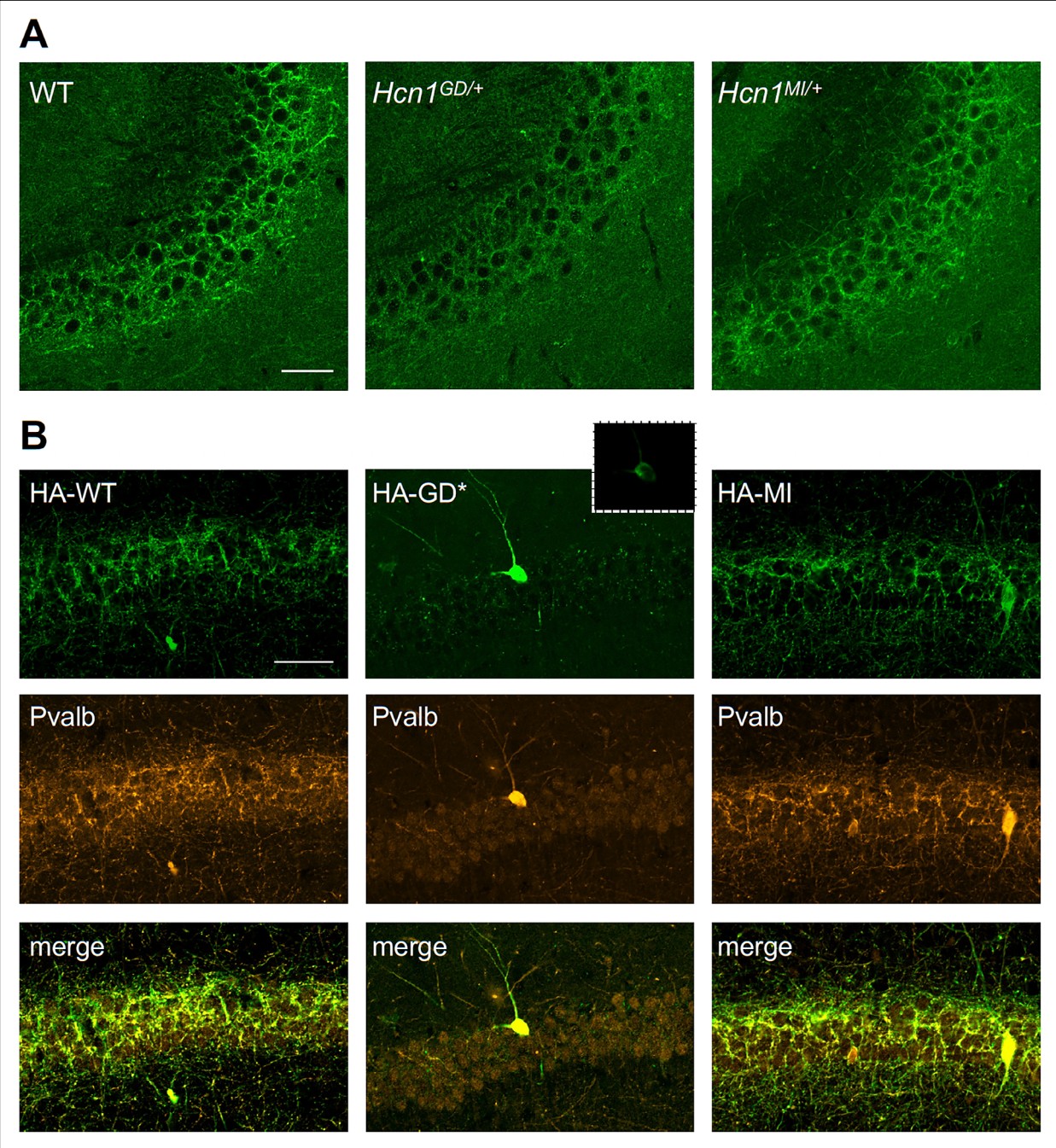

**Figure 7.** Viral targeting of HCN1 subunits to parvalbumin-positive (PV+) interneuron terminals in hippocampus shows impaired trafficking of HCN1-GD mutant protein. Immunofluorescent staining of midcoronal sections from adult mouse brain. (**A**) Labeling of HCN1 protein in the hippocampus of WT, *Hcn1GD/+*, and *Hcn1MI/+* mice. Images show a close-up of the pyramidal cell layer of area CA3, with the PV+ interneuron axonal terminals visible. Scale bar = 50 µm for all panels. (**B**) Anti-hemagglutinin (anti-HA) tag labeling of virally expressed HCN1 protein after stereotaxic injection into hippocampal area CA1 of adult *Pvalb-Cre* mice. Top panels show distribution of HA-HCN1-WT (HA-WT), HA-HCN1-GD (HA-GD), and HA-HCN1-MI (HA-MI) protein at 21 days after injection. Equivalent laser power and gain were used for images in HA-WT, HA-MI, and inset for HA-GD sample; the HA-GD* full panel image was obtained at higher laser power and gain to compensate for the lower expression levels of the HA-GD protein. Middle row panels show co-labeling for parvalbumin protein, with merged images shown in bottom panels. Note that HA-tagged protein labeling is visible in the soma and dendrites of PV+ interneurons in all samples, presumably owing to virus overexpression (see *Figure 7—figure supplements 1 and 2*). However, only HA-WT and HA-MI are also visible in axonal terminals, with minimal or absent labeling of axons noted in HA-GD samples even when laser power and gain are increased. Scale bar = 50 µm for all panels.

The online version of this article includes the following figure supplement(s) for figure 7:

*Figure 7 continued on next page*

**Figure supplement 1.** Impaired trafficking of virally transduced HCN1-GD protein to parvalbumin-positive (PV+) interneuron terminals in hippocampus.

**Figure supplement 2.** Viral targeting of HCN1 subunits to parvalbumin-positive (PV+) interneurons in hippocampal area CA1.

that disruption of PV+ interneuron function, due to HCN1 protein misfolding or altered trafficking, may represent an important component of neuronal circuit dysfunction in HCN1-linked epilepsy.

## Paradoxical response of $Hcn1^{GD/+}$ and $Hcn1^{MI/+}$ mice to antiseizure medications

Clinical reports from at least four different patients with *HCN1*-linked DEE (carrying variants p.M305L, p.G391D, and p.I380F; *Marini et al., 2018*; *Bleakley et al., 2021*; C. Marini, personal communication, March 2021) consistently revealed worsening of seizures after administration of either lamotrigine (LTG) or phenytoin. A particularly striking example is provided by the p.G391D variant, identified in two unrelated patients with similar course of disease. Both patients had neonatal seizure onset, characterized by daily asymmetric tonic seizures with apnea and cyanosis, severe developmental delay, and died between 14 and 15 months of age. Both patients also showed a paradoxical response to phenytoin, which caused the induction of status epilepticus. Given this precedent, we hypothesized that there may be systematic similarities between the adverse pharmacological response profile of patients with p.G391D and p.M153I variants and $Hcn1^{GD/+}$ and $Hcn1^{MI/+}$ mice, which would further endorse their use as models with construct, face, and potentially predictive validity.

Consistent with our hypothesis, we found that administration of LTG (23 mg/kg, i.p.) induced convulsive seizures, defined as grade 3 or higher on a modified Racine scale (see 'Materials and Methods'), within 90 min from injection in 11/15 $HCN1^{GD/+}$ animals (*Figure 8A and C*, left). Several animals had multiple seizure bouts, ranging from rearing with forelimb clonus to wild running and jumping seizures, during the period of observation. Video ECoG recordings were performed on a subset of animals (2/15 mutant and 2 WT control animals) for a total of 24 hr after drug injection, confirming the presence of GTCS-like activity on ECoG in both animals (*Figure 8A*) but no seizure activity in control littermates (data not shown). Similar outcomes were seen in $Hcn1^{MI/+}$ animals, where convulsive seizures were observed in 6/9 mice in response to LTG (*Figure 8A and C*, right). Of note, four of the six $Hcn1^{MI/+}$ animals with adverse response had grade 6 seizures, again higher than the maximum seizure grade observed in $Hcn1^{GD/+}$ mice (*Figure 8C*). Video ECoG recordings performed on three $Hcn1^{MI/+}$ animals confirmed the presence of GTCS electrographic activity in 2/3 animals (*Figure 8A*, right). In both lines, all seizure events recorded through video ECoG were reflected in behavioral manifestations, implying that data collected through observation in non-implanted animals are likely to have captured all seizure events generated in response to LTG injection.

Results with phenytoin (30 mg/kg PE, i.p.) were even more striking as 14/18 animals tested from the $Hcn1^{GD/+}$ line developed convulsive seizures within 2.5 hr of injection (this number includes 11/14 animals with no prior drug exposure and 3/4 animals previously exposed to LTG; *Figure 8B and C*, left). Latency to first seizure after injection was longer than observed with LTG, in line with the slower time to peak plasma concentration for phenytoin. In addition, multiple seizures were observed during a period up to 6 hr after injection, again consistent with the longer half-life of phenytoin compared to LTG (*Markowitz et al., 2010*; *Hawkins et al., 2017*). Among $Hcn1^{MI/+}$ animals, we observed seizures in 10/15 mice following phenytoin injection, with similar temporal pattern and ECoG signatures compared to $Hcn1^{GD/+}$ mice (this number includes 6/8 animals without prior drug exposure and 4/7 animals previously exposed to LTG, *Figure 8B and C*, right).

All the $Hcn1^{GD/+}$ or $Hcn1^{MI/+}$ animals tested, either in response to LTG or phenytoin, were randomized to receive a control vehicle injection either 1 week before or 1 week after the drug administration session. None had convulsive seizures in response to vehicle injection during the period of observation (4 hr for behavioral experiments and 24 hr for video ECoG experiments), further confirming that the seizures observed in the period following drug administration were a direct consequence of the pharmacological challenge (*Figure 8C*). As an additional control, we tested the effects of a third anticonvulsant, namely, sodium valproate (VPA), which has not been reported to worsen seizures in any of the patients with pathogenic *HCN1* sequence variations in our database. Administration of sodium valproate (250 mg/kg, i.p.) did not result in the occurrence of convulsive seizures in any of the

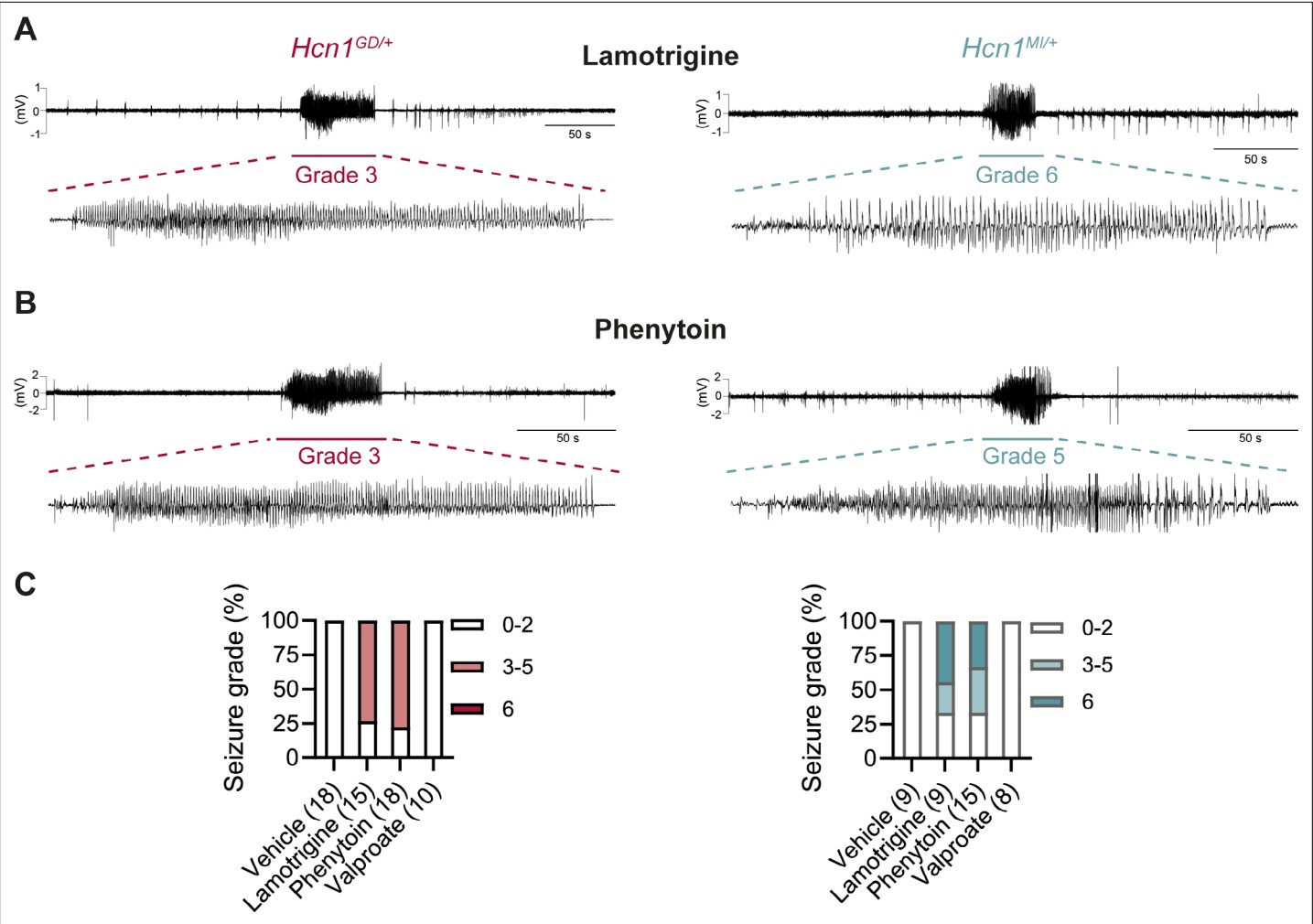

**Figure 8.** Anticonvulsant-drug induced seizures in *Hcn1GD/+* and *Hcn1MI/+* mice. (**A**) Example electrocorticogram (ECoG) trace for lamotrigine-induced grade 3 seizure in *Hcn1GD/+* (left) and grade 6 seizure in *Hcn1MI/+* mouse (right), with seizure shown at expanded time scale below. (**B**) Phenytoin-induced grade 3 seizure in *Hcn1GD/+* (left) and grade 5 seizure in *Hcn1MI/+* mouse (right). (**C**) Percentage of *Hcn1GD/+* (left) and *Hcn1MI/+* (right) mice experiencing convulsive seizures (grade 3 and higher) upon administration of vehicle, lamotrigine, phenytoin, or sodium valproate, with highest seizure grade indicated by color. As in the case of spontaneous seizures, *Hcn1MI/+* animals showed an overall higher seizure severity in response to drug administration. No convulsive seizures were observed in response to vehicle or sodium valproate during the observation period of the experiment (see 'Materials and methods'). Total number of animals tested for each condition is reported in parentheses (see 'Results' for the number of video ECoG recordings versus behavioral observations).

*Hcn1GD/+* or *Hcn1MI/+* animals tested (*Figure 8C*). However, use of 250 mg/kg VPA resulted in profound sedation of *Hcn1GD/+* mice, which was not seen in *Hcn1MI/+* mice or WT littermate controls, with 9/10 *Hcn1GD/+* animals failing to respond to gentle touch or passive moving of the tail 45 min after drug injection versus 2/8 *Hcn1MI/+* and 0/8 WT animals (note that tail pinching still elicited a response in all mice tested).

What may be the mechanism by which LTG and phenytoin induce seizures in mice with pathogenic *HCN1* variants? Although both drugs act as Na+ channel blockers, LTG is also widely considered an HCN channel activator, a property thought to contribute to its antiepileptic effects (*Poolos et al., 2002*). Despite multiple reports on its action on $I_h$ in neurons, showing somewhat inconsistent outcomes (*Peng et al., 2010*; *Huang et al., 2016*), no studies have been conducted to directly test the effect of LTG on isolated HCN channels expressed in heterologous systems. To obtain further clarification on the mechanism of action of LTG, we tested the drug on wildtype HCN1 and HCN2 channels after transient transfection in HEK293T cells (*Figure 9*). Surprisingly, we found that LTG had no effects on either the midpoint voltage of activation or maximal tail current density of either HCN1

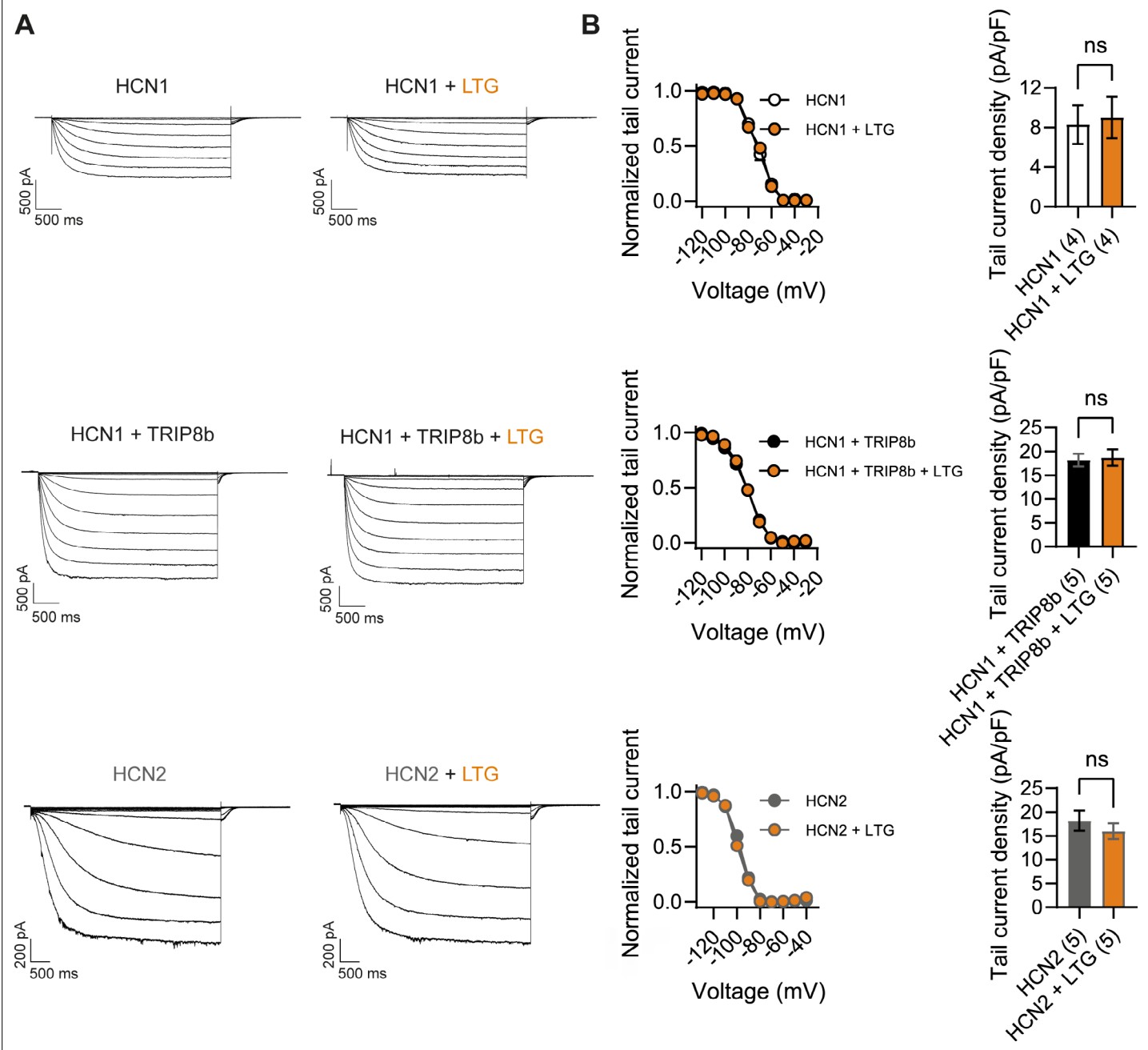

**Figure 9.** Lamotrigine (LTG) has no direct effect on HCN1 or HCN2 channel activity. (**A**) Sample current traces from whole-cell voltage-clamp recordings in HEK293T cells transiently expressing HCN1 (top), HCN1 and TRIP8b (middle) or HCN2 (bottom), in the absence or presence of bath applied LTG (100 µM). Voltage step protocol for HCN1 and HCN1 + TRIP8b: holding potential was –20 mV (1 s), with steps from –30 mV to –120 mV (–10 mV increments, 3.5 s) and tail currents recorded at –40 mV (3.5 s); for HCN2: holding potential was similarly –20 mV (1 s), but steps were applied from –40 mV to –130 mV in –15 mV increments (5 s) and tail currents recorded at –40 mV (5 s). (**B**) Tail current activation curves (left) and tail current density (right) group data obtained for each condition. Lines show data fitting with a Boltzmann function (see 'Materials and methods'). Data show that neither the midpoint voltage of activation ($V_{1/2}$) nor the maximum tail current density was altered by LTG addition (ns = not significant, paired *t*-test; see **Figure 9—source data 1** for numerical values). Data in (**B**) represent mean ± SEM (note that smaller error bars in the tail current activation curves may be obscured by the circles representing mean values).

The online version of this article includes the following source data and figure supplement(s) for figure 9:

**Source data 1.** Lamotrigine has no direct effect on HCN1 or HCN2 channel activity.

**Figure supplement 1.** Effects of lamotrigine (LTG) on Na$_V$1.5 channels.

or HCN2 channels; LTG also had no effects when HCN1 was coexpressed with its auxiliary subunit TRIP8b (*Figure 9B*). In contrast, under the same conditions, LTG exerted its expected inhibitory effect on Na$_v$1.5 sodium channel currents (*Figure 9—figure supplement 1*; *Qiao et al., 2014*). These results clearly show that LTG is not a direct activator of HCN channels and suggest that any effects reported in neurons are the result of indirect regulation of $I_h$, perhaps secondary to Na$^+$ channel block, and likely contingent on cell-specific, state-dependent signal transduction pathways. Although further experiments in mouse brain slices are needed to fully rule out an effect of this drug on native HCN channels with pathogenic sequence variants, it seems likely the effects of LTG on *Hcn1$^{GD/+}$* and *Hcn1$^{MI/+}$* animals are due to its action to directly block Na$^+$ channels – similar to the effects of phenytoin – rather than an enhancement of HCN currents.

While we have at present no available indication that either of the two patients carrying the *HCN1* p.M153I variant showed adverse effects to LTG or phenytoin treatment, our results in mice suggest that despite the clear differences in their phenotypes, there may be some fundamental similarities in the disease mechanisms at play both in patients and mouse models for *HCN1*-linked DEE.

## Reduced efficacy of HCN/I$_h$-blocking compounds on HCN1-G391D channels and *Hcn1$^{GD/+}$* neurons

The observation that several of the known *HCN1* variants, when tested for their properties in vitro (*Nava et al., 2014*; *Marini et al., 2018*; *Porro et al., 2021*) or in vivo (*Bleakley et al., 2021*), exhibit an increase in 'leak' current suggests that use of HCN channel blockers may be beneficial in the treatment of patients with certain *HCN1* sequence variants. This seems especially warranted as the 'leak' component has been directly linked to hyperexcitability in cortical pyramidal neurons (*Bleakley et al., 2021*). Several such blockers exist, although at present none is able to efficiently cross the blood–brain barrier. More importantly, all currently available inhibitors that are specific for HCN channels, such as ZD7288, ivabradine, zatebradine, cilobradine, and their derivatives (*Melchiorre et al., 2010*; *Del Lungo et al., 2012*), are thought to act through the same mechanism, namely, as pore blockers by interacting with residues located within the pore cavity. At the same time, many of the most severe *HCN1* variants, including p.G391D, are located in the S6 transmembrane domain of the channel, which forms the inner lining of the pore. Such observations raise the question whether pathogenic mutations affecting pore structure may decrease the efficacy of currently available HCN-blocking compounds.

The structural models presented in *Figure 10F* illustrate the proximity of the G391D mutation to the known binding site of ZD7288 in the HCN pore cavity, wherein the residue immediately preceding G391 (V390 in human HCN1, corresponding to V379 in mouse) was shown to critically affect the sensitivity of the channel to the drug (*Cheng et al., 2007*). The presence of charged aspartate residues in G391D-containing subunits leads to a disruption in the symmetry of the pore in heteromeric channels containing at least one WT subunit, and consequently in the relative distance between the V390 side chains facing the cavity. To test whether this predicted disruption in the ZD7288 binding site leads to a reduced efficacy of the drug, we expressed heteromeric HCN1 channels containing WT and G391D subunits in HEK cells. As illustrated in *Figure 10A and B*, compared to the effect of ZD7288 to reduce the current expressed by WT HCN1 channels by ~67%, the drug was minimally effective in blocking either the instantaneous or the time-dependent current component generated by mutant WT/G391D channels (~15% inhibition). Addition of Cs$^+$, another HCN channel blocker (albeit a less specific one, compared to ZD7288) with a different mechanism of action and binding site (*DiFrancesco, 1982*), led to a strong reduction in the current expressed by both WT and G391D channels (*Figure 10A*), confirming that the observed currents were indeed generated by HCN1 WT/G391D heteromeric channels.

Parallel experiments in brain slices, where bath application of ZD7288 is the most common method used to isolate the contribution of $I_h$ to neuronal physiology, demonstrate the additional difficulties posed by the limited efficacy of the drug in *Hcn1$^{GD/+}$* neurons. As shown in *Figure 10C–E*, application of the compound did not modify either the RMP or input resistance in *Hcn1$^{GD/+}$* neurons, while appropriately eliminating voltage sag, hyperpolarizing the RMP, and increasing input resistance in WT neurons. These results leave unanswered the question whether the positive shift in RMP in *Hcn1$^{GD/+}$* neurons is directly due to 'leaky' HCN1 channels or is a secondary response to changes in other conductances. The development of new HCN-blocking compounds with different mechanisms of action will be important both in allowing a more targeted investigation of the role of the 'leak' component in

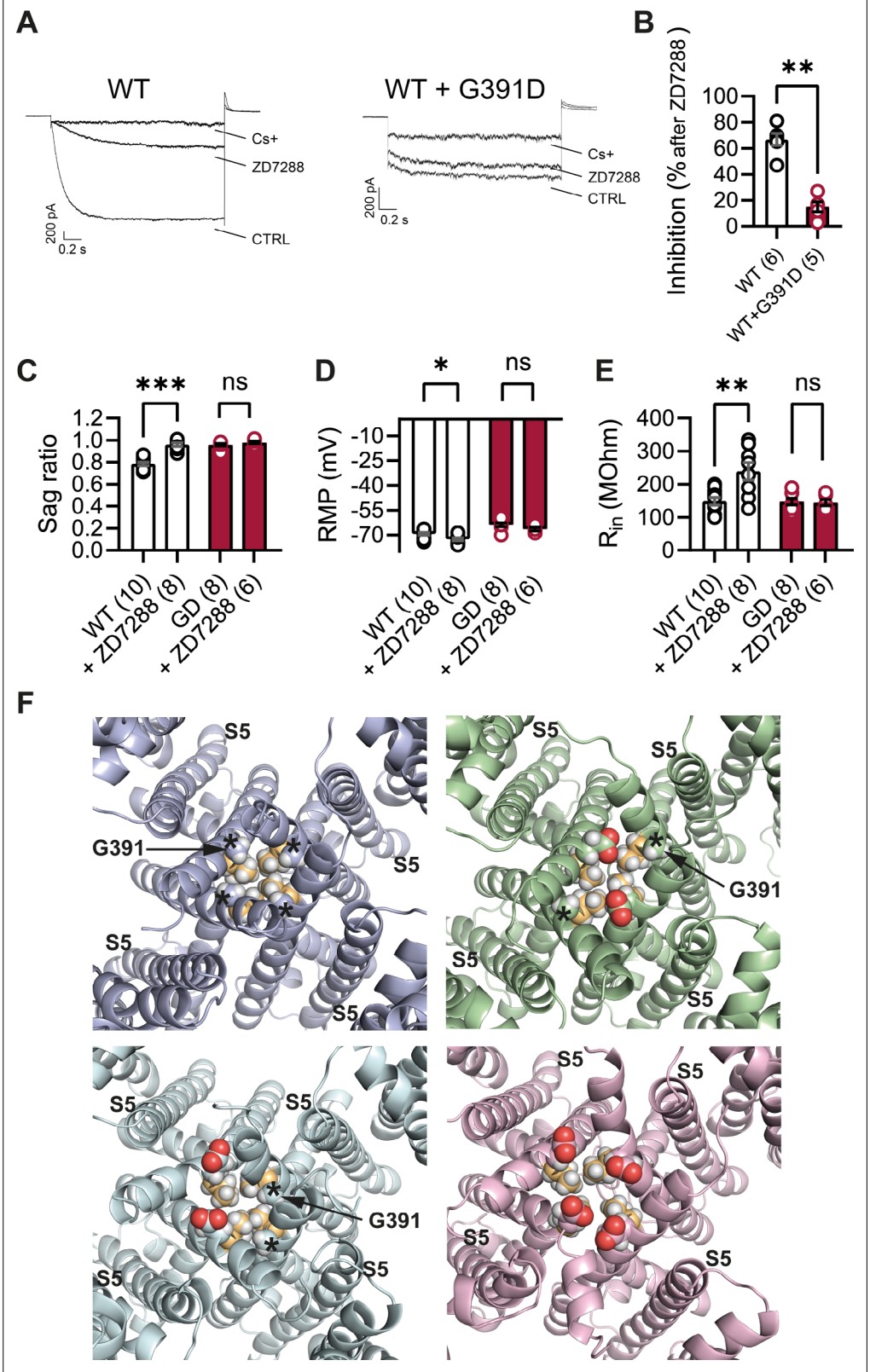

**Figure 10.** Lack of effect of ZD7288 on *Hcn1*$^{GD/+}$ neurons and heterologously expressed hHCN1-WT/G391D channels. (**A**) Whole-cell voltage-clamp recordings in HEK293T cells heterologously expressing either human HCN1 (hHCN1) WT or heteromeric hHCN1 WT/G391D channels. Sample current traces are shown, in response to a voltage step to –110 mV from a holding potential of –20 mV, before (control, CTRL) and after full block from

*Figure 10 continued on next page*

*Figure 10 continued*

ZD7288 (30 µM) or CsCl (5 mM) bath application. (**B**) Bar plot with % inhibition from group data (WT = 66.83 ± 4.5% versus WT/G391D = 15.00 ± 3.89%; ***p<0.001, Mann–Whitney *U* test). (**C–E**) The HCN blocker ZD7288 (10 µM) abolished sag (**C**), significantly hyperpolarized resting membrane potential (RMP) (**D**), and increased input resistance (**E**) in WT, but did not alter any of these parameters *Hcn1GD/+* neurons (**p<0.01, ***p<0.001, paired *t*-test; number of recorded cells is indicated in the bar graphs; number of animals used: WT n = 4, WT + ZD7288 n = 3, *Hcn1GD/+* n = 3, *Hcn1GD/+* + ZD7288 n = 3; see *Figure 10—source data 1* for numerical values). (**F**) Snapshots of the human HCN1 pore region structure at the end of a 100 ns molecular dynamics (MD) simulation run, with or without introduction of the G391D mutation in two or four subunits. Top left: WT; top right: heteromer with G391D variant on two opposite subunits; bottom left: heteromer with G391D variant on two adjacent subunits; bottom right: homomer with G391D variant on all four subunits. The outer pore helix S5 is labeled (S5), with asterisks indicating the position of the G391 residue in all cases. Carbonyls of modeled G391D residues are colored in red. Side chains of V390 residues are colored in light orange. Data in (**B–E**) represent mean ± SEM.

The online version of this article includes the following source data for figure 10:

**Source data 1.** Lack of effect of ZD7288 on *Hcn1GD/+* neurons.

## Discussion

The speed and precision of next-generation sequencing (NGS) allow the efficient genetic assessment of children that present with early-onset epilepsies. The overarching goal of such precision medicine is to identify the specific molecular and network mechanisms that link distinct epilepsy variants in individual genes to particular disease profiles, resulting in more effective gene- or variant-specific therapeutic strategies. Despite the fact that dysfunction in ion channels carrying missense mutations can be promptly modeled in heterologous expression systems, such assessments do not sufficiently reflect the dysfunction caused by the sequence variations in vivo, preventing the direct translation of such datasets into applicable clinical strategies. In our study, we set out to fill this information gap by designing new preclinical models, which reproduce key disease features of *HCN1*-linked DEE. The *Hcn1GD/+* and *Hcn1MI/+* knock-in mice we generated display not only an appropriate and robust epilepsy phenotype, along with comorbid behavioral abnormalities, but illustrate the difficulties intrinsic to the treatment of genetic developmental epilepsies, including the adverse response of individuals with *HCN1* variants to conventional antiseizure medications, and the potential for altered drug sensitivity of mutant channels. These results underscore how mutations in voltage-gated ion channels can fundamentally alter network properties and the response of neuronal circuits to pharmacological challenge. Some of these alterations may be acute, while others may result from HCN1 channel dysfunction impacting brain development, as multiple studies have demonstrated the influence of HCN channels on cortical maturation (*Schlusche et al., 2021*) and the complex interplay between HCN isoforms in shaping developing neuronal networks (*Surges et al., 2006*; *Bender and Baram, 2008*; *Stoenica et al., 2013*). In this scenario, our models provide a useful system for testing mechanistic hypotheses and developing new therapeutic approaches.

A notable feature of our models and patients with *HCN1* sequence variations is the paradoxical induction of seizures by LTG and phenytoin. A similar phenomenon has been reported both in Dravet syndrome patients carrying *SCN1A* pathogenic variants (*Perucca and Perucca, 2019*) and in mice modeling *Scn1a* haploinsufficiency (*Scn1a+/-*; *Hawkins et al., 2017*). In Dravet syndrome, loss-of-function mutations in the *SCN1A* sodium channel gene are thought to promote seizures by exerting a net effect to reduce the excitability of inhibitory interneurons (*Catterall, 2018*). This has led to the suggestion that certain anticonvulsant drugs acting as Na$^+$ channel blockers paradoxically exacerbate seizures in this syndrome as a result of a synergistic action to further suppress inhibition. Interestingly, the axon-specific expression of HCN1 channels in PV+ interneurons matches the similarly polarized subcellular distribution of voltage-gated Na$^+$ channels, wherein Na$_V$1.1 (encoded by *SCN1A*) is the predominant subtype expressed in the axon of PV+ interneurons (*Ogiwara et al., 2007*; *Ogiwara et al., 2013*; *Dutton et al., 2013*; *Hedrich et al., 2014*; *Hu and Jonas, 2014*). To maintain ionic homeostasis during repetitive firing, PV+ interneuron axons also strongly express Na$^+$/K$^+$-ATPases

(*Peng et al., 1997*), which will activate in response to Na⁺ entry, leading to membrane hyperpolarization due to electrogenic pump activity. Recent pharmacological studies in vitro have proposed that axonal expression of HCN channels can counteract the hyperpolarizing Na⁺ pump current during repetitive firing, allowing for fast AP firing and propagation (*Roth and Hu, 2020*). Thus, blockade of HCN channel activity in PV+ interneuron axon terminals results in a failure in AP propagation and reduced perisomatic inhibition onto target excitatory neurons (*Southan et al., 2000*; *Aponte et al., 2006*; *Roth and Hu, 2020*). Since the only HCN subtype expressed in PV+ interneuron terminals is HCN1, diminished GABAergic input onto excitatory neurons as a result of HCN1 protein dysfunction could contribute to excitation/inhibition imbalance and epilepsy. Such an effect could be exacerbated by further inhibition of Na⁺ channel activity by antiseizure medications, similar to what is thought to occur in Dravet syndrome. HCN1 subunits also substantially contribute to regulating the excitability of a second class of inhibitory neurons, namely, somatostatin-positive interneurons, which target the dendrites of pyramidal cells (*Matt et al., 2011*). Here, HCN1 acts along with HCN2 to set the somatic resting potential and facilitate the spontaneous activity of this class of interneurons.

While our histological analysis provides some evidence as to the deleterious effects of the HCN1-GD variant on axonal function, there are some important limitations to the interpretation of this data in the absence of a direct physiological evaluation of the intrinsic and synaptic properties of PV+ interneurons. Future studies aimed at investigating the physiological effects of HCN1 mutations in genetically identified interneuron populations, along with conditional knock-in mouse lines that limit *Hcn1^GD* or *Hcn1^MI* allele expression to select classes of inhibitory neurons, will provide a more definitive answer to the question of how different variants may affect perisomatic and/or dendritic inhibition in *HCN1*-linked epilepsy.

When tested in heterologous expression systems, epilepsy-linked HCN1 mutations have been shown to affect the channel in several different ways. These actions include loss of function due to decreased HCN1 channel activation, decreased protein levels, and/or impaired targeting to the plasma membrane (*Porro et al., 2021*). Some of the variants exert a dominant negative effect when a mutant subunit assembles into a heteromeric channel with WT subunits. Conversely, gain-of-function effects include accelerated activation kinetics, depolarized midpoint voltage of activation (as in M153I), and an increased voltage-independent component of the HCN current, that is, generation of a depolarizing 'leak' current (as in G391D). Why do *HCN1* mutations that in heterologous expression systems have divergent effects on channel function equally result in epilepsy, and a similar paradoxical response to certain Na⁺ channel blockers? Our results in *Hcn1^GD/+* and *Hcn1^MI/+* mice show that both these variants ultimately led to a net decrease in $I_h$-dependent voltage sag, and a significant decrease in protein expression levels (*Figure 5*), despite the p.M153I variant yielding an apparent gain-of-function channel upon heterologous expression (*Marini et al., 2018*). This suggests that misfolding and impaired biosynthesis of mutant HCN1 subunits may be a prevalent consequence likely to affect the function of neurons in multiple ways. An increase in cellular stress, for example, would be expected to affect PV+ neurons in particular, as they are known to be extremely sensitive to metabolic and oxidative stress (*Kann, 2016*). Such a mechanism may explain why two variants with seemingly divergent biophysical alterations in channel functional properties ultimately cause overlapping phenotypes.

The divergent characteristics of the two variants do, however, lead to distinguishable phenotypes. This is demonstrated by the overall greater severity of the *Hcn1^GD/+* phenotype (reduced brain size, altered gait, impaired pole climbing, lack of spontaneous alternation, and greater reductions in HCN1 protein and AP firing), as well as the remarkable sex dependence of mortality in *Hcn1^MI/+* mice. This observation suggests that, while some fundamental mechanisms are affected in similar ways, there are distinctions in the manner in which physiological networks are perturbed by different *HCN1* variants. Moreover, similar to other ion channel-linked syndromes, a variety of disease phenotypes are observed in patients carrying *HCN1* pathogenic variants, with DEEs seen in ~20% of cases (*Marini et al., 2018*). Overall, if both familial and sporadic patients are considered, the predominant phenotypes are milder and include genetic generalized epilepsies (*Bonzanni et al., 2018*), as well as the spectrum of genetic epilepsy with febrile seizures plus (GEFS+) (*Marini et al., 2018*). In addition, some variants do not show full penetrance. An analogous picture emerges from the study of a third *HCN1* variant recently modeled in mice (*Hcn1^M294L*, corresponding to human p.Met305Leu; *Bleakley et al., 2021*). Similar to HCN1-G391D, this variant generates a channel with a significant 'leak' or

voltage-independent current component, although protein levels in the $Hcn1^{M294L}$ mouse brain are less dramatically affected. The epileptic phenotype of $Hcn1^{M294L}$ mice is considerably milder than either $Hcn1^{GD/+}$ or $Hcn1^{MI/+}$ animals. Intriguingly, however, $Hcn1^{M294L}$ mice are also adversely affected by LTG, which caused the induction of seizures, similar to heterozygous animals from the two lines presented here. From the therapeutic point of view, the observed mechanistic convergence does offer some hope that patients with genetic HCN1 channel dysfunction can be treated following similar principles or strategies across different variants. A larger convergence in treatment strategies, centered on the degree of interneuron dysfunction across multiple epilepsies, may also be envisioned to emerge in the future.

Finally, what may such strategies look like? Our data once again stress the difficulty of devising targeted therapeutic approaches for ion channel-associated DEE syndromes using existing tools. As stated above, several of the identified epilepsy-associated *HCN1* variants generate channels with what have been deemed 'gain of aberrant' function properties (*Bleakley et al., 2021*), implying that HCN channel blockers may be potentially helpful. At the same time, several of the most severe pathogenic *HCN1* variants are located in the S6 segment, which forms the lining of the pore cavity, or in the immediate vicinity to the channel's intracellular gate. These include G391D (*Figure 10*) and two more variants at the same position, G391C and G391S, as well as M379R, I380F, A387S, F389S, I397L, S399P, and D401H (*Nava et al., 2014*; *Lucariello et al., 2016*; *Marini et al., 2018*; *Wang et al., 2019*). While we do not have complete information available about the effects of known HCN inhibitors on the other variants, results presented here using ZD7288 on G391D-containing HCN1 channels strongly suggest that the mutations may decrease the efficacy of the pore-blocking compounds, reducing their ability to suppress both the time-dependent and time-independent current component generated by the mutant channels (*Marini et al., 2018*; *Porro et al., 2021*). There is therefore a need to develop new small-molecule compounds that target the HCN1 channels at sites outside the pore cavity.

The available 3D structures of HCN1, combined with molecular dynamics simulations, coarse-grained modeling, and molecular docking, suggest that there are many opportunities for such a pharmacological-targeting approach to succeed (*Lee and MacKinnon, 2017*; *Gross et al., 2018*; *Porro et al., 2019*). The structures have indeed revealed unique intracellular regulatory modules, including both the HCN domain and C-linker/CNBD, which may allow for the targeting of HCN channels through allosteric modulation of pore gating properties. Furthermore, the recently determined structures of HCN4, the main cardiac isoform of HCN channels, have demonstrated differences in the relative arrangement of these regulatory modules (*Saponaro et al., 2021*), which may be exploited towards the development of subtype-specific drugs. Compound library screenings for small molecules with HCN1-isoform specificity have already yielded some initial and promising results (*McClure et al., 2011*; *Harde et al., 2019*). Other approaches include the design of cell-penetrant peptides, which interfere with the interaction between HCN channels and their regulatory auxiliary subunit, TRIP8b (*Han et al., 2015*; *Saponaro et al., 2018*), as well as antisense oligonucleotide-based strategies aimed at HCN1 mRNA downregulation. Of note, the latter strategy may be particularly effective in light of the hypothesis that protein misfolding may be an important component of the pathology in *HCN1*-linked DEE, together with the observation that *Hcn1* null mice do not have spontaneous seizures (*Huang et al., 2009*; *Santoro et al., 2010*).

In conclusion, the availability of strong preclinical models, such as the $Hcn1^{GD/+}$ and $Hcn1^{MI/+}$ mouse lines presented here, provides an ideal platform for further investigation of mechanisms underlying epileptic encephalopathies and testing new and paradigm-shifting therapeutic strategies.

## Materials and methods

### Key resources table

| Reagent type (species) or resource | Designation | Source or reference | Identifiers | Additional information |
|---|---|---|---|---|
| Antibody | Anti-HCN1 (mouse monoclonal) | NeuroMab | RRID:AB_2877279; clone N70-28, Cat# N75-110 | IHC (1:300) |
| Antibody | Anti-NPY (rabbit polyclonal) | Immunostar Hudson | RRID:AB_572253; Cat# 22940 | IHC (1:1000) |

*Continued on next page*

*Continued*

| Reagent type (species) or resource | Designation | Source or reference | Identifiers | Additional information |
|---|---|---|---|---|
| Antibody | Anti-GFAP (mouse monoclonal) | Invitrogen | RRID:AB_10598206; clone GA5, Cat# 14-9898-82 | IHC (1:250) |
| Antibody | Anti-ATP1a3 (mouse monoclonal) | Invitrogen | RRID:AB_2274447; clone G10, Cat# MA3-915 | IHC (1:400) |
| Antibody | Anti-HA tag (rat monoclonal) | Roche | RRID:AB_390917; clone BMG-3F10, Cat# 12013819001 | IHC (1:100) |
| Antibody | Anti-$K_v$1.2 (mouse monoclonal) | NeuroMab | RRID:AB_2296313; clone K14/16, Cat# N75-008 | IHC (1:300) |
| Antibody | Anti-parvalbumin (rabbit polyclonal) | Synaptic Systems | RRID:AB_2156474; Cat# 195002 | IHC (1:1000) |
| Cell line (human) | HEK293T | ATCC (authenticated by STR profiling) | RRID:CVCL_0063 | Tested for mycoplasma: negative result |
| Commercial assay, kit | Clariom S Assay, mouse | Thermo Fisher Scientific | Cat# 902931 | |
| Genetic reagent (*Mus musculus*) | $Hcn1^{-/-}$ (B6.129S-$Hcn1^{tm2Kndl}$/J) | Jackson Laboratory | RRID:IMSR_JAX:016566 | Males and females |
| Genetic reagent (*M. musculus*) | $Hcn1^{G380D}$ (C57BL/6J-$Hcn1^{em1(G380D)Cecad}$) | This paper | | Males and females |
| Genetic reagent (*M. musculus*) | $Hcn1^{M142I}$ (C57BL/6J-$Hcn1^{em2(M142I)Cecad}$) | This paper | | Males and females |
| Genetic reagent (*M. musculus*) | Pvalb-Cre (B6.129P2-$Pvalb^{tm1(cre)Arbr}$/J) | Jackson Laboratory | RRID:IMSR_JAX:017320 | Males and females |
| Peptide, recombinant protein | WFA biotin conjugate | Sigma-Aldrich | RRID:AB_2620171; Cat# L-1516 | IHC (1:1000) |
| Peptide, recombinant protein | Streptavidin Alexa Fluor 594 conjugate | Life Technologies | RRID:AB_2337250; Cat# S32356 | IHC (1:500) |
| Recombinant DNA reagent | pcDNA 3.1 (plasmid) | Invitrogen | | Used for HEK293T experiments to express human HCN1 and mouse TRIP8b |
| Recombinant DNA reagent | pCI (plasmid) | Promega | U47119; Cat# E1731 | Used for HEK293T experiments to express mouse HCN2 |
| Recombinant DNA reagent | pmaxGFP (plasmid) | Amaxa Biosystems | | HEK293T experiments (cotransfection for target cell visualization) |
| Recombinant DNA reagent | pIRES-EGFP (plasmid) | Clontech Laboratories | | Used for HEK293T experiments to express human $Na_v$1.5 |
| Recombinant DNA reagent | pAAV-hSyn-DIO-hM4D(Gi)-mCherry (plasmid) | Addgene | Cat# 44362 | Used to replace hM4D(Gi)-mCherry with mouse HCN1 cDNAs |
| Recombinant DNA reagent | pAAV-hSyn-DIO-HA-HCN1-WT (plasmid) | This paper | | Mouse HCN1 with N-terminal HA tag (YPYDVPDYA); AAV virus injection |
| Recombinant DNA reagent | pAAV-hSyn-DIO-HA-HCN1-GD (plasmid) | This paper | | Mouse HCN1-GD with N-terminal HA tag (YPYDVPDYA); AAV virus injection |
| Recombinant DNA reagent | pAAV-hSyn-DIO-HA-HCN1-MI (plasmid) | This paper | | mouse HCN1-MI with N-terminal HA tag (YPYDVPDYA); AAV virus injection |
| Sequence-based reagent | crRNA for $Hcn1^{GD}$ | This paper | | 5'-ACTGGATCAA AGCTGTGGCA-3' |

*Continued on next page*

*Continued*

| Reagent type (species) or resource | Designation | Source or reference | Identifiers | Additional information |
|---|---|---|---|---|
| Sequence-based reagent | ssODN for *Hcn1*$^{GD}$ | This paper | | 5'-CCCAAGCCCCTGTCAGCATGTCTGACCTCTGG ATTACCATGCTGAGC ATGATTGTGGGCGCCACCTGCTACGCAATGTTTGTT GATCATGCCACAG CTTTGATCCAGTCTTTGGACTCTTCAAGGAG-3' |
| Sequence-based reagent | crRNA for *Hcn1*$^{MI}$ | This paper | | 5'-ATCATGCTTAT AATGATGGT-3' |
| Sequence-based reagent | ssODN for *Hcn1*$^{MI}$ | This paper | | 5'-ATCGGATGCCA CGTTGAAAATAATCCACGGTGTTGTCGTCTGCTCTG TGAAGAAC GTGATTCCAACTGGTATGATGACCAAATTTCCAACG ATCATTATAA GCATGATTAAATCCCAATAAAACCTA-3' |
| Sequence-based reagent | *Hcn1*$^{GD}$ WT forward primer | This paper | | 5'-ACGGTGATGA CACTTGTTCAGT-3' |
| Sequence-based reagent | *Hcn1*$^{GD}$ WT reverse primer | This paper | | 5'-TGGATCAAAGCTGTGGCATGGC-3' |
| Sequence-based reagent | *Hcn1*$^{GD}$ mutant forward primer | This paper | | 5'-ACCTGCTACGC AATGTTTGTTGAT-3' |
| Sequence-based reagent | *Hcn1*$^{GD}$ mutant reverse primer | This paper | | 5'-GGCACTACA CGCTAGGAA-3' |
| Sequence-based reagent | *Hcn1*$^{MI}$ forward primer | This paper | | 5'-CAACATTTGTT TGTTCTCCTCACC-3' |
| Sequence-based reagent | *Hcn1*$^{MI}$ reverse primer | This paper | | 5'-ATGATCGAAT GCCACGTTGA-3' |
| Software, algorithm | ImageJ | NIH | | v1.49 |
| Software, algorithm | Axograph X | Axograph Scientific | | 1.7.6 |
| Software, algorithm | MATLAB | MathWorks | | R2022a |
| Software, algorithm | Prism | GraphPad | | v9.0.1 |
| Software, algorithm | OriginPro | OriginLab | | v2016 |
| Software, algorithm | pClamp 10 | Molecular Devices | RRID:SCR_011323 | v10.7 |
| Software, algorithm | EZ Patch | Elements srl | | v 1.2.3 |
| Software, algorithm | TAC4.0 | Thermo Fisher Scientific | | v4.0.2.15 |
| Transfected construct (human) | HCN1 (cDNA) | Xention Ltd (Cambridge, UK) | | HEK293T experiments |
| Transfected construct (human) | Na$_V$1.5 (cDNA) | PMID:33213388 | | HEK293T experiments |
| Transfected construct (mouse) | HCN1 (cDNA) | PMID:11331358 | | Used to generate AAV virus construct and mutants thereof (see 'Materials and methods') |
| Transfected construct (mouse) | HCN2 (cDNA) | PMID:11331358 | | HEK293T experiments |
| Transfected construct (mouse) | TRIP8b (cDNA) | PMID:19555649 | | HEK293T experiments |

## Animals

Mouse colonies were maintained both at the University of Cologne and at Columbia University in New York. For animals housed in Cologne, mice were kept in type II long plastic cages under standard housing conditions (21 ± 2°C, 50% relative humidity, food ssniff Spezialitäten GmbH, Soest, Germany) and water ad libitum; individually ventilated cages or ventilated cabinets (SCANBUR) and an inverted 12:12 dark:light cycle (with light turning on at 10 pm). All experiments were in accordance with European, national, and institutional guidelines and approved by the State Office of North Rhine-Westphalia, Department of Nature, Environment and Consumer Protection (LANUV NRW, Germany). For animals housed in New York, mice were maintained on a 12 hr light–dark cycle (with light turning on at 7 am) under standard housing conditions as above, with ad libitum access to food (Pico Lab rodent diet 5053 for general maintenance or Pico Lab mouse diet 5058 for breeders; Lab Diet, St. Louis, MO) and water. Weanlings were supplemented with DietGel Recovery (ClearH2O, Westbrook, ME) for 2 weeks post-weaning. All animal experiments were conducted in accordance with policies of the NIH Guide for the Care and Use of Laboratory Animals and the Institutional Animal Care and Use Committee (IACUC) of Columbia University.

The following commercially available mouse lines were used: B6.129P2-*Pvalb*[tm1(cre)Arbr]/J (Jackson Laboratories stock 017320; Bar Harbor, ME) and B6.129S-*Hcn1*[tm2Kndl]/J (Jackson Laboratories, stock 016566). All animals were maintained on a C57BL/6J background.

## Mouse genome editing

We employed CRISPR/Cas9 genome editing using the Easy Electroporation of Zygotes (EEZy) approach in order to generate *Hcn1*[G380D] (C57BL/6J-*Hcn1*[em1(G380D)Cecad], hereafter referred to as *Hcn1*[GD]) and *Hcn1*[M142I] (C57BL/6J-*Hcn1*[em2(M142I)Cecad], hereafter referred to as *Hcn1*[MI]) mice as previously described (**Tröder et al., 2018**). gRNAs were selected using CRISPOR (**Haeussler et al., 2016**). *Hcn1*[GD] mice were generated using the crRNA sequence 5'-ACTGGATCAAAGCTGTGGCA-3' and the ssODN sequence 5'-CCCAAGCCCCTGTCAGCATGTCTGACCTCTGGATTACCATGCTGAGCATGATTGTGGGCGCC ACCTGCTACGCAATGTTTGTTGATCATGCCACAGCTTTGATCCAGTCTTTGGACTCTTCAAGGAG-3', containing a new Bcl1 restriction site. *Hcn1*[MI] mice were generated using the crRNA sequence 5'-ATCATGCTTATAATGATGGT- 3' and the ssODN sequence 5'-ATCGGATGCCACGTTGAAAATAATCC ACGGTGTTGTCGTCTGCTCTGTGAAGAACGTGATTCCAACTGGTATGATGACCAAATTTCCAAC GATCATTATAAGCATGATTAAATCCCAATAAAACCTA-3', containing a new DpnII restriction site. Custom crRNAs (Alt-R), generic tracrRNAs, and ssODN (Ultramers) were purchased from Integrated DNA Technologies (Coralville, IA). Genome editing was performed in C57BL/6J mice at the in vivo Research Facility of the CECAD Research Center, University of Cologne, Germany. Identical procedures were used to generate *Hcn1*[GD] and *Hcn1*[MI] mice in the C57BL/6J background at the Genetically Modified Mouse Models shared resource of Columbia University, New York. All founder lines (two for *Hcn1*[GD] and two for *Hcn1*[MI]) were assessed for basic phenotypic features (growth, brain size, presence of spontaneous seizures, pathohistological markers, HCN1 protein expression, drug response) with no differences noted.

## Mouse genotyping

DNA was isolated from ear biopsies in 100 µl lysis buffer (100 mM NaCl, 50 mM Tris/HCl pH 8.0, 1 mM EDTA, 0.2% Nonidet P-40, 0.2% Tween 20, 0.1 mg/ml Proteinase K) overnight at 54°C under constant shaking, followed by 40 min at 84°C to inactivate proteinase K. Amplification was performed in PCR reaction mix using a touchdown PCR protocol with the following primers: *Hcn1*[GD] wildtype (WT) forward 5'- ACGGTGATGACACTTGTTCAGT-3', reverse 5'-TGGATCAAAGCTGTGGCATGGC -3' (468 bp); *Hcn1*[GD] mutant forward 5'-ACCTGCTACGCAATGTTTGTTGAT-3', reverse 5'-GGCACTACAC GCTAGGAA-3' (354 bp). For *Hcn1*[MI] genotyping, the following primers flanking the mutation site were used: forward 5'-CAACATTTGTTTGTTCTCCTCACC-3', reverse 5'-ATGATCGAATGCCACGTTGA-3' (248 bp), and a restriction digest followed to identify heterozygous animals containing the DpnII restriction site. DNA purification was performed with GeneJET PCR Purification kit (Thermo Scientific, Germany) according to the manufacturer's instructions. Enzyme restriction with DpnII was performed in 30 µl reaction mix (1 µl DpnII, 3 µl DpnII buffer, 10 µl purified DNA, and 16 µl H2O; New England Biolabs, Ipswich, MA) and incubation at 37°C for 2 hr. Gel electrophoresis was performed with 1.8% agarose gels (VWR Life Science, Sigma-Aldrich) in 1× TAE buffer (40 mM Tris, 10 mM acetic acid, 1 mM

EDTA pH 8.0) using a 200 bp DNA ladder (PANladder, PAN-Biotech). Genotyping for each animal was performed in two technical replicates at the beginning and end of the experiments, respectively.

## ECoG recordings
### Surgery
Adult animals of both genotypes and sexes were implanted with radio transmitters (PhysioTel ETA-F10, Data Sciences International) for long-term ECoG and video monitoring. For analgesia, mice received 0.025 mg/kg buprenorphine (i.p., TEMGESIC, Indivior, UK Limited) prior to surgery and 5.0 mg/kg Carprofen (s.c., Norbrook Laboratories Limited, Ireland) during surgery. Mice were anesthetized with 0.5–4% isoflurane in 100% oxygen and kept at 0.8–1.5% isoflurane throughout the surgery. Body temperature was maintained at 36.5°C using a homeothermic heating pad (Stoelting, Germany). Mice were placed into a stereotaxic device (Kopf Instruments, CA), a midline skin incision was made above the skull, and the periosteum was denatured by a short treatment with 10% $H_2O_2$, followed by rinsing with 0.9% NaCl solution, drying of the skull, and application of dental cement (OptiBond, Kerr Dental, Germany) to cover the skull surface. The transmitter body was implanted subcutaneously in a pouch made in the loose skin of the back. The two lead wires were tunneled subcutaneously through the incision on the skull. Using a dental drill, a small hole was drilled above the hippocampus (AP –2.0, ML +2.0 in millimeters from Bregma, always on the right side) for the recording wire, and another hole was made above the cerebellum for the reference wire. The wires were placed into the holes so as to touch the dura and fixed with dental cement. The skin was subsequently closed with tissue glue (GLUture, World Precision Instruments, WPI, USA) and animals were allowed to recover for 5 days before data acquisition.

### Data acquisition and analysis
Recordings were performed by placing the animal's home cage onto a receiver board that detected the ECoG and movement activity signals, and together with the synchronized video recordings data were digitally stored using Ponemah (DSI, MN) and the Media Recorder (Noldus, Wageningen, The Netherlands). Seizure and spike detection was performed with the software NeuroScore (DSI) using an integrated, automated spike detection tool (absolute threshold: threshold value 200 µV, maximum value 2000 µV; spike duration: 0.1–250 ms) and manually verified by the experimenter. Graphical visualization of the seizures was performed with a custom script in MATLAB (MathWorks, MA). MATLAB code used to generate *Figure 4A and C* can be found in *Source code 1*.

## Behavioral analysis
All experiments were performed during the dark cycle when animals are naturally active.

### Behavioral grading of seizures
For evaluation of seizures, animals were monitored electrographically and behaviorally and seizures graded using a modified Racine scale (*Van Erum et al., 2019*), with 0 = normal behavior; 1 = mouth and facial movements; 2 = head nodding; 3 = forelimb clonus; 4 = rearing with forelimb clonus, falling; 5 = wild running, jumping; 6 = laying on the side with fore- and hindlimb clonus. When multiple classes of severity occurred during one electrographically defined seizure, the most severe behavior represented the grade of that seizure.

### Open field
The open field was performed in a box (50 × 50 × 40 cm) illuminated with 100 lux. Mice were placed in one corner of the arena facing the walls and could freely move for 15 min. Tracks were recorded and analyzed with the software EthoVision XT 16 (Noldus). The following parameters were obtained: distance moved, running velocity, number of rotations, time in border (an imaginary 5 cm wide border around the arena), and time in center (an imaginary inner square of 20 × 20 cm).

### Gait analysis
For the automated gait analysis system Catwalk XT (Noldus), mice had to run on an enclosed walkway on a glass plate. Run duration variation was set to 0.5–20 s, with a maximum speed variation of 60%,

and a minimum of 10 consecutive steps. A minimum of eight runs per mouse was collected. Runs were classified with the Catwalk XT 10 software (Noldus). Footprints were detected automatically by the software and manually corrected by visual inspection. The following parameters were used for analysis: running speed, stand (duration in seconds of contact of a paw with the glass plate), stride length (distance between successive placements of the same paw in centimeter), step cycle (the time in seconds between two consecutive paw placements), BOS (the average width between the front or hind paws), step sequence (contains information on the order in which four paws are placed), and regularity index (expresses the number of normal step sequence patterns relative to the total number of paw placements in percent).

## Pole test

Mice were placed head upward on the top of a vertical wooden rod (60 cm long, 7 mm diameter) so that they grasp the rod with all four paws and had to climb down the pole in a headfirst position, as described (*Freitag et al., 2003*). For motivation to climb down, nest material was placed at the bottom of the pole. The ability to turn the body 180° was evaluated, and the time needed to reach the floor with all four paws was assessed. A trial was considered successful if the mouse climbed down after a 180° body turn at levels 1 or 2 (i.e., the top and middle portion of the pole), and not successful if the turn occurred at the bottom level 3 (*Figure 2G*) or when the mice were sliding down or falling. To test motor learning, mice were scored three times with an intertrial interval of 30 s, during which they were placed in their home cage. Maximum trial duration was 120 s.

## Spontaneous alternation

The test was performed over 2 days using a Y-shaped maze consisting of three equally sized arms (34 × 5 × 30 cm) made of transparent Plexiglas. Mice were placed in the center of the maze and allowed to freely explore until they performed 24 transitions or after a maximum trial duration of 15 min (*Kitanaka et al., 2015*). An entry into any arm was considered a transition. Alternation behavior was defined as consecutive entries into all three arms without repeated entries and was expressed as percentage of the total arm entries. As an example, if the three arms are labeled A, B, and C, and the mouse enters the arms in the following sequence – ABCBACABCACBABCBACBC – then the total alternation opportunities would be 18 (total entries minus 2); the actual behavioral alternations would be 12; and the percent alternation would be 66.7% (*Kitanaka et al., 2015*). The alternation rate was thus calculated as (number of alternations/(number of transitions – 2)) × 100%, and the mean of both days was used for comparisons between groups.

## Object recognition memory

The test was performed in a box (50 × 50 × 40 cm) illuminated with 100 lux. For orientation, distal landmarks were placed outside the arena and attached to the surrounding walls. Prior to training, the mice were placed in the box and allowed to habituate for 5 min. Then, 1 hr later, during training, the mice were allowed to freely explore the box in the presence of two identical objects (blue cooling bottles) until they reached a total object interaction time of 20 s or for a maximum trial duration of 15 min. Mice that explored the objects for less than 20 s during the 15 min training were excluded. For testing, 24 hr later the mice were again placed in the box with two objects, but this time one object was replaced with a novel object (magic cube), whereas the location remained the same as during training. The determination of which object was replaced was randomized and balanced. Tracks were recorded and analyzed with the software EthoVision XT 16 (Noldus). Object interaction was quantified as the amount of time the mouse's nose was within 2 cm around the object. The relative exploration times were expressed as a discrimination index (D.I. = $(t_{novel} - t_{familiar})/(t_{novel} + t_{familiar})$).

## Drug testing

LTG (23 mg/kg lamotrigine isethionate; Tocris, Minneapolis, MN) and valproate (250 mg/kg sodium valproate; Tocris) were dissolved in saline (0.9% NaCl). Phenytoin (30 mg/kg Phenhydan; Desitin, Hamburg, Germany) was diluted in 21% cyclodextran/1× phosphate-buffered saline (PBS) titrated to pH 8.0 or administered as fosphenytoin (45 mg/kg fosphenytoin sodium, corresponding to 30 mg/kg phenytoin equivalents or PE; Millipore-Sigma, St. Louis, MO) dissolved in saline. Saline and 21% cyclodextran/1× PBS, respectively, were used as vehicle. All drugs were applied by intraperitoneal injection

(i.p.). For behavioral seizure assessment, animals were observed for a total of 4 hr after injection. For video ECoG seizure assessment, animals were recorded for 24 hr after injection. The highest severity seizure displayed during the observation period was scored as the seizure grade for that animal's response to drug administration.

## Immunohistochemistry

Animals were perfused with 1× PBS followed by 4% paraformaldehyde in 1× PBS, and brains post-fixed overnight at 4°C. After several washes in 1× PBS, 40 μm coronal slices were cut using a vibratome, and free floating sections permeabilized in PBS + 0.1% Triton, followed by incubation in blocking solution (1× PBS + 5% normal donkey serum) for 1 hr at room temperature. Primary antibody incubation was carried out in blocking solution overnight at 4°C. Antibodies used were mouse monoclonal anti-HCN1 (clone N70-28, NeuroMab 75-110, dilution 1:300; Davis, CA); rabbit polyclonal anti-NPY (ImmunoStar 22940, dilution 1:1000; Hudson, WI); mouse monoclonal anti-GFAP (clone GA5, Invitrogen 14-9892-82, dilution 1:250); mouse monoclonal anti-ATP1a3 (clone G10, Invitrogen MA3-915, dilution 1:400); rat monoclonal anti-HA tag (clone BMG-3F10, Roche 12013819001, dilution 1:100); mouse monoclonal anti-$K_V$1.2 (clone K14/16, NeuroMab 75-008, dilution 1:300); and rabbit polyclonal anti-parvalbumin (Synaptic Systems 195002, dilution 1:1000). Secondary antibody incubation was performed in blocking solution for 2 hr at room temperature. All secondary antibodies were used at 1:500 dilutions: goat anti-mouse IgG1 cross-adsorbed (Alexa Fluor 488, Life Technologies A21121; Eugene, OR), donkey anti-rabbit cross-adsorbed (Alexa Fluor 488, Life Technologies A21206), goat anti-rabbit Superclonal (Alexa Fluor 647, Life Technologies A27040), goat anti-rat IgG (H+L) cross-adsorbed (Alexa Fluor 488, Life Technologies, A11006); and goat anti-mouse IgG2b cross-adsorbed (Alexa Fluor 488, Life Technologies, A21141). For perineuronal net staining, WFA biotin conjugate was used (Sigma L1516, dilution 1:1000), followed by incubation with Streptavidin (Alexa Fluor 594 conjugate, Life Technologies S32356, dilution 1:500). Nissl stain was performed by adding NeuroTrace reagent (NT 640/660, Thermo Fisher N21483, dilution 1:500) during the secondary antibody incubation, followed by extensive washing in 1× PBS. Images were acquired on a Zeiss LSM 700 laser scanning confocal microscope with Zen 2012 SP5 FP3 black edition software, using either a Zeiss Fluar ×5/0.25 objective (0.5 zoom, pixel size: 2.5 × 2.5 μm$^2$) or a Zeiss Plan-Apochromat ×20/0.8 objective (1.0 zoom, pixel size: 0.3126 × 0.3126 μm$^2$).

Image analysis and fluorescent signal quantification were performed using ImageJ 1.49v software (National Institutes of Health, USA). For viral injection evaluations, signal intensity was quantified in 2–4 slices per injection, located within 240 μm from the target site and avoiding any damaged areas as identified by Nissl counterstain. Regions of interest (ROIs) were drawn using the parvalbumin co-labeling to identify the CA1 pyramidal layer and circumvent any PV+ cell bodies (see *Figure 7—figure supplement 1*) so as to avoid introducing noise due to the random distribution of labeled somas in the various samples. Similarly sized ROI areas were drawn for each image, and mean intensity within the area was measured in arbitrary units (au).

## Stereotaxic virus injection

Mice were anesthetized using isoflurane (Covetrus, Portland, ME) and provided analgesics with 5 mg/kg Carprofen s.c. (Zoetis, Troy Hills, NJ). A craniotomy was performed above the target region and a glass pipette was stereotaxically lowered to the desired depth. Injections were performed using a nano-inject II apparatus (Drummond Scientific), with 25 nl of solution delivered every 15 s until a total amount of 200 nl was reached. The pipette was retracted after 5 min. Viruses were injected bilaterally, at a titer of 2 × 10$^{12}$ vg/ml, with injection coordinates AP –2.0, ML ±1.5, DV –1.4 (in millimeters from Bregma). One single virus injection was performed per hemisphere, and each hemisphere counted as an independent injection site.

For virus construction, Addgene plasmid #44362 (pAAV-hSyn-DIO-hM4D(Gi)-mCherry) was cut with restriction enzymes NheI and AscI, and the hM4D(Gi)-mCherry cDNA was replaced with mouse HCN1 cDNA including an N-terminal HA tag (YPYDVPDYA) immediately following the starting methionine. Site-directed mutagenesis was performed by Applied Biological Materials Inc (Richmond, BC, Canada) and each viral DNA packaged into AAV serotype 8 and serotype 9 by the Duke University Viral Vector Core (Durham, NC). Experiments performed with either virus serotype produced analogous results.

## Slice electrophysiology

### Slice preparation

Mice were anesthetized by inhalation of isoflurane (5%) for 7 min, subjected to cardiac perfusion of ice-cold carbogenated artificial cerebrospinal fluid, modified for dissections (d-ACSF; 195 mM sucrose, 10 mM glucose, 10 mM NaCl, 7 mM $MgCl_2$, 0.5 mM $CaCl_2$, 25 mM $NaHCO_3$, 2.5 mM KCl, 1.25 mM $NaH_2PO_4$, 2 mM Na-pyruvate, pH 7.2) for 30 s before decapitation according to the procedures approved by the IACUC of Columbia University. The skull was opened, and the brain removed and immediately transferred into ice-cold carbogenated d-ACSF. The hippocampus was dissected in both hemispheres. Each hippocampus was placed in the groove of an agar block and 400-µm-thick hippocampal slices were cut using a vibrating tissue slicer (VT 1200, Leica, Germany) and transferred to a chamber containing a carbogenated mixture of 50% d-ACSF and 50% ACSF (22.5 mM glucose, 125 mM NaCl, 1 mM $MgCl_2$, 2 mM $CaCl_2$, 25 mM $NaHCO_3$, 2.5 mM KCl, 1.25 mM $NaH_2PO_4$, 3 mM Na-pyruvate, 1 mM ascorbic acid, pH 7.2) at 35°C, where they were incubated for 40–60 min. Thereafter, slices were held at room temperature (21°C) until transfer into the recording chamber.

### Electrophysiology

Slices were transferred from the incubation chamber into the recording chamber of an Olympus BX51WI microscope (Olympus, Japan), where they were held in place by a 1 mm grid of nylon strings on a platinum frame. Slices were continually perfused with ACSF at 34 ± 1°C, maintained by a thermostat-controlled flow-through heater (Warner Instruments, CT, USA). Healthy somas of CA1 pyramidal neurons were identified visually under ×40 (20 × 2) magnification and patched under visual guidance using borosilicate glass pipettes (I.D. 0.75 mm, O.D. 1.5 mm, Sutter Instruments, UK) with a tip resistance of 4–5.5 MΩ, connected to a Multiclamp 700B amplifier (Molecular Devices, CA) and filled with intracellular solution, containing (in mM): 135 K-gluconate, 5 KCl, 0.1 EGTA, 10 HEPES, 2 NaCl, 5 MgATP, 0.4 $Na_2GTP$, 10 $Na_2$-Phosphocreatin, adjusted to a pH of 7.2 with KOH. Recordings were only accepted if the series resistance after establishing a whole-cell configuration did not exceed 25 MΩ and did not change by more than 20% of the initial value during the course of the experiment.

### Pharmacology

Stock solution of 10 mM ZD7288 (Tocris, UK) was stored at –20°C and diluted in ACSF to a concentration of 10 µM before bath application to the slice. In some experiments, blockers of $GABA_A$ and $GABA_B$ receptors, 2 µM SR95531 and 2 µM CGP55845 (Tocris, UK), respectively, were added to the bath solution. As application of the latter did not alter any of the reported measures and since synaptic transmission was not assessed in this case, data were pooled between experiments performed with and without blockers of inhibition.

### Data acquisition and analysis

Electrophysiological recordings were digitized, using a Digidata 1322 A A/D interface (Molecular Devices), at a sampling rate of 20 kHz (low-pass filtered at 10 kHz) and recorded with the pClamp 10 software (Molecular Devices). The amplifier settings of the Multiclamp 700B were controlled through Multiclamp Commander (Molecular Devices).

After 50 ms baseline recording, 1 s current steps of –350 to +350 pA were applied to the patched cells in increments of 25 pA, after which an additional 1 s of post-step membrane potential was recorded. The trigger time between these episodes was 3 s. Voltage deflections in response to current steps of –50 to +50 pA were used to calculate the input resistance. Initial RMP was obtained immediately upon breaking into the cell. Voltage sag in response to negative current steps was calculated by dividing the steady-state voltage deflection during the late phase of the –100 pA current step by the peak of the voltage deflection during the same step. AP threshold was determined as the membrane voltage at which the derivative of the voltage trace exceeded 40 mV/ms. AP width describes the width of the AP at 50% of its amplitude.

Data was analyzed using Axograph X software (Axograph Scientific, Australia), MATLAB (MathWorks), as well as Microsoft Excel (Microsoft Corp., WA) and Prism 8 (GraphPad, CA) and visualized in Acrobat Illustrator (Adobe, CA).

## HEK293 cell electrophysiology

### Constructs

The cDNAs encoding full-length human HCN1 channel and mouse TRIP8b (splicing variant 1a-4) were cloned into the pcDNA 3.1 (Invitrogen) mammalian expression vector. The cDNA encoding the full-length mouse HCN2 was cloned in pCI (Promega) mammalian expression vector. The cDNA encoding the full-length human $Na_V1.5$ channel was cloned in pIRES-EGFP (Clontech Laboratories) mammalian expression vector.

### Cell culture and transfection

HEK293T cells were cultured in Dulbecco's modified Eagle's medium (Euroclone) supplemented with 10% fetal bovine serum (Euroclone), 1% Pen Strep (100 U/ml of penicillin and 100 µg/ml of strepto-mycin), and grown at 37°C with 5% $CO_2$. When ~70% confluent, HEK293T cells were transiently trans-fected with cDNA using Turbofect transfection reagent (Thermo Fisher Scientific, Germany) according to the manufacturer's recommended protocol. For each 35 mm Petri dish, 1 µg of the plasmid DNA and 0.3 µg of EGFP-containing vector (pmaxGFP, Amaxa Biosystems) were used. For HCN1 and TRIP8b coexpression, 1 µg of each cDNA was used in combination with 0.3 µg of EGFP-containing vector.

### Chemicals

Lamotrigine isethionate (Tocris) and ZD7288 (Tocris) stock solutions were prepared by dissolving the powders in Milli-Q water to obtain a final stock concentration of 100 mM and 10 mM, respectively. Single-use aliquots were made and stored at −20°C until the day of the experiment.

### Electrophysiology and data analysis

30–72 hr after transfection, the cells were dispersed by trypsin treatment. Green fluorescent cells were selected for patch-clamp experiments at room temperature (about 25°C). Currents were recorded in whole-cell configuration either with a ePatch amplifier (Elements, Cesena, Italy) or with a Dagan3900A amplifier (Dagan Corporation, MN); data acquired with the Dagan3900A amplifier were digitized with an Axon Digidata 1550B (Molecular Devices) converter. All data were analyzed offline with Axon pClamp 10.7. Patch pipettes were fabricated from 1.5 mm O.D. and 0.86 mm I.D. borosilicate glass capillaries (Sutter, Novato, CA) with a P-97 Flaming/Brown Micropipette Puller (Sutter) and had resis-tances of 3– 6 MΩ. For the recordings of HCN channels, the pipettes were filled with a solution containing 10 mM NaCl, 130 mM KCl, 1 mM egtazic acid (EGTA), 0.5 mM $MgCl_2$, 2 mM ATP (magne-sium salt), and 5 mM HEPES–KOH buffer (pH 7.2), while the extracellular bath solution contained 110 mM NaCl, 30 mM KCl, 1.8 mM $CaCl_2$, 0.5 mM $MgCl_2$, and 5 mM HEPES–KOH buffer (pH 7.4). For the recordings of $Na_V1.5$ channel, the pipettes were filled with a solution containing 5 mM NaCl, 140 mM CsCl, 4 mM ATP (magnesium salt), 2 mM $MgCl_2$, 5 mM EGTA, 10 mM HEPES–CsOH buffer (pH 7.4) while the extracellular bath solution contained 135 mM NaCl, 4 mM KCl, 1 mM $CaCl_2$, 2 mM $MgCl_2$, 20 mM D-glucose, 10 mM HEPES–NaOH buffer (pH 7.4). To assess HCN channel activation curves, different voltage-clamp protocols were applied depending on the HCN subtype: for HCN1 and HCN1 coexpressed with TRIP8b, holding potential was –20 mV (1 s), with steps from –30 mV to –120 mV (–10 mV increments, 3.5 s) and tail currents recorded at –40 mV (3.5 s); for HCN2, holding potential was –20 mV (1 s), with steps from –40 mV to –130 mV (–15 mV increments, 5 s) and tail currents recorded at –40 mV (5 s). Patch-clamp currents were acquired with a sampling rate of 5 kHz and low-pass filtered at 2.5 kHz. $Na_V1.5$ recordings were obtained from holding potentials of –80 mV or –130 mV, as indicated. Currents were elicited by stepping to –30 mV for 50 ms followed by a step to the next holding potential (–60 or –130 mV for 5 s) in a cycling manner. To measure the inactivation curves, a voltage step protocol was used starting from a holding potential of –100 mV for 150 ms followed by a series of inactivating pulses (from –30 to –130 mV, for 500 ms); the fraction of channels that remain available after each inactivating pulse were assessed by the peak currents during the following short test pulse at 0 mV for 50 ms. LTG was applied to the (extracellular) bath solution of the experiments at 100 µM concentration. For HCN channels, the effect was assessed after 20 min of application of LTG in the bath solution. The percentage of ZD7288 induced block (30 µM concentra-tion, bath applied) was assessed by measuring the current amplitude at the end of the hyperpolarizing pulse at –110 mV, which includes both the instantaneous and the time-dependent component of the

current. After reaching the full effect of ZD7288, a 5 mM CsCl-containing solution was added to the bath to verify current response to a second known HCN channel blocker. Mean activation curves were obtained by fitting maximal tail current amplitude, plotted against the voltage step applied, with the Boltzmann equation: $y = 1/[1 + \exp((V–V_{1/2})/k)]$, where V is voltage, y the fractional activation, $V_{1/2}$ the half-activation voltage, and k the inverse-slope factor = $–RT/zF$ (all in mV), using OriginPro software (OriginLab, Northampton, MA). Mean $V_{1/2}$ values were obtained by fitting individual curves from each cell to the Boltzmann equation and then averaging all the obtained values. All measurements were performed at room temperature.

## Differential gene expression analysis

RNA was prepared from whole hippocampus (using material from both hemispheres per animal) from 6-month-old male mice (n = 4 per group; for WT samples, we used two WT littermates from the $Hcn1^{GD}$ line and two WT littermates from the $Hcn1^{MI}$ line) according to the manufacturer's instructions. Briefly, tissue from hippocampus (~40 mg) was submerged in RNAprotect Tissue Reagent (QIAGEN, Germany). Subsequent RNA isolation was performed with the RNeasy Mini Kit (QIAGEN) and RNase-free DNase Set (QIAGEN). RNA concentration was determined by UV absorbance using an Epoch Microplate spectrophotometer (BioTek, USA). RNA integrity was assessed using an automated gel electrophoresis system (Agilent 4200 Tape Station System, Agilent Technologies, CA). RNA quality was evaluated by assessing RNA integrity number (samples' RIN range: 8.7–9.0) and optical density values at 260/280 nm (samples' ratio range: 1.96–2.12). cRNA synthesis, labeling, fragmentation, array hybridization, washing, and staining, as well as microarray scanning and signal processing were performed by the Gene Expression Affymetrix Facility (Center for Molecular Medicine Cologne). Briefly, 100 ng total RNA were used for reverse transcription and the resulting cDNA was fragmented and labeled via the GeneChip WT PLUS Reagent Kit as per the manufacturer's instructions (Affymetrix, Thermo Fisher Scientific, USA). The labeled cDNA samples were hybridized to Affymetrix Clariom S Mouse arrays (Thermo Fisher Scientific, Cat# 902931) and incubated in GeneChip Hybridization Oven-645 (Affymetrix, Thermo Fisher Scientific) rotating at 60 rpm at 45°C for 16 hr. Subsequently, arrays were washed on a GeneChip Fluidics Station-450 (Affymetrix, Thermo Fisher Scientific) and stained with the Affymetrix HWS kit according to the manufacturer's protocol. Finally, the chips were scanned with an Affymetrix Gene-Chip Scanner-3000-7G and the Affymetrix GCOS software was used for the generation of .DAT and .CEL files. Data analysis was performed with the Transcriptome Analysis Console software (TAC version 4.0.2, Thermo Fisher Scientific), using SST-RMA for summarization, and a LIMMA Bioconductor package for differential gene expression evaluation with empirical Bayesian (eBayes) analysis and Benjamini–Hochberg method for false discovery rate (FDR)-adjusted p-value calculations. Unfiltered results for 112 voltage-gated ion channel gene candidates were extracted and ranked in ascending order of raw p-value, independent of fold-change or FDR threshold. The candidate list comprised $K_V$1-4, $K_V$7, $K_V$10-12, $K_{2P}$1-18, $K_{ir}$1-7, $Na_V$1.1–1.9, $Ca_V$1-3, TPR Group 1, HCN, $K_{Ca}$ and $K_{Na}$ channels; beta, gamma, delta, and other modulatory subunits, as well as 'silent' subunits in the $K_V$5,6,8,9 subfamilies, CNG channels, Group 2 TRP channels, TPC, and CatSper channels were not included. All .CEL and .CHP files used for this analysis were deposited in the NCBI GEO database (accession GSE209630).

## Data analysis

Sample size estimation was based on prior studies and experience (*Festing, 2018*). For behavioral experiments in *Figure 2*, sample size was calculated using a power analysis assuming an effect size of 0.5, a power of 0.8, and an α error of 0.05, based on previous experience (G*Power 3.1.9.2, University Düsseldorf, http://www.gpower.hhu.de). No data were excluded after analysis. Statistical analysis was performed using GraphPad Prism (version 9.0.1, GraphPad). Parameters were assessed for normality using the D'Agostino & Pearson test. For normally distributed data, means were compared using a two-tailed Student's *t*-tests (assuming equal variances between genotypes) and paired data was analyzed with a paired *t*-test. For repeated measurements, a two-way repeated-measures (RM) ANOVA was performed, and, when appropriate, a post hoc Šídák's multiple-comparisons test followed. For nonparametric data, medians were compared using the Mann–Whitney *U* test, paired data were analyzed with a Wilcoxon matched-pairs signed-rank test, and in case of two grouping factors a Kruskal–Wallis test with post hoc Dunn's multiple comparisons, when appropriate, was performed. All

tests were two-tailed, and statistical significance accepted at $p < 0.05$. All parameters were assessed for a sex difference with a mixed-effects analysis, having *genotype* and *sex* as grouping factors, and in case of significant sex effects data from female and male mice were plotted and analyzed separately. Unless otherwise stated, data represent mean ± standard error of the mean (SEM).

## Acknowledgements

We acknowledge the support of animal facilities for excellent mouse care Cologne: Esther Mahabir-Brenner (CMMC team), Maria Guschlbauer (Medical Faculty team), Branko Zevnik (CECAD in vivo Research Facility); New York: Christine Winnicker (ZMBBI). We are thankful for the expert assistance of the Transgenic Core Unit of the CECAD in vivo Research Facility (Prof. Branko Zevnik) and the Genetically Modified Mouse Model Shared Resource (GMMMSR) at Columbia University (Dr. Chyuan-Sheng Victor Lin) for the generation of transgenic mice. We are also grateful to the Gene Expression Affymetrix Facility (CMMC, Cologne; Prof. Agapios Sachinidis) and the Biomedical Informatics Shared Resource (Columbia University; Prof. Richard A Friedman) for assistance in the generation and analysis of Affymetrix datasets. We thank Daniel Bauer for providing PDB files and advice on *HCN1* variant structural modeling, and Chris Reid, Nick Poolos, Carla Marini, and Wayne Frankel for helpful discussion during the course of the study. This work was supported by grants from the German Research Foundation (DFG, FOR 2715) (IS63/10-1/2) and CRC 1451 (project ID 431549029-B01) to DI; Telethon award GGP20021 to AMo; NIH grants NS106983, NS109366, and NS123648 to SAS; NIH CCSG grant NCI 5P30CA013696-44 and the Columbia Precision Medicine Initiative for the generation of mouse models of human disease.

## Additional information

### Funding

| Funder | Grant reference number | Author |
|---|---|---|
| National Institutes of Health | NS106983 | Steven A Siegelbaum |
| National Institutes of Health | NS109366 | Steven A Siegelbaum |
| National Institutes of Health | NS123648 | Steven A Siegelbaum |
| Deutsche Forschungsgemeinschaft | FOR 2715 (IS63/10-1/2) | Dirk Isbrandt |
| Deutsche Forschungsgemeinschaft | CRC 1451 (project ID 431549029 - B01) | Dirk Isbrandt |
| Fondazione Telethon | GGP20021 | Anna Moroni |

The funders had no role in study design, data collection and interpretation, or the decision to submit the work for publication.

### Author contributions

Andrea Merseburg, Conceptualization, Funding acquisition, Investigation, Project administration, Supervision, Validation, Visualization, Writing – review and editing; Jacquelin Kasemir, Investigation, Visualization, Writing – review and editing; Eric W Buss, Data curation, Investigation, Validation; Felix Leroy, Tobias Bock, Data curation, Investigation, Validation, Writing – review and editing; Alessandro Porro, Data curation, Investigation, Validation, Visualization, Writing – review and editing; Anastasia Barnett, Data curation, Investigation; Simon E Tröder, Methodology, Resources, Validation, Writing – review and editing; Birgit Engeland, Investigation, Project administration, Supervision, Validation; Malte Stockebrand, Investigation, Project administration, Validation; Anna Moroni, Steven A Siegelbaum, Conceptualization, Funding acquisition, Project administration, Resources, Supervision, Writing – review and editing; Dirk Isbrandt, Conceptualization, Data curation, Funding acquisition, Project administration, Resources, Supervision, Writing – review and editing; Bina Santoro, Conceptualization,

Funding acquisition, Investigation, Project administration, Supervision, Validation, Visualization, Writing – original draft, Writing – review and editing

### Author ORCIDs
Andrea Merseburg  http://orcid.org/0000-0003-0630-6564
Jacquelin Kasemir  http://orcid.org/0000-0001-9176-5241
Eric W Buss  http://orcid.org/0000-0003-0473-4717
Felix Leroy  http://orcid.org/0000-0003-1715-3233
Tobias Bock  http://orcid.org/0000-0002-7734-1183
Alessandro Porro  http://orcid.org/0000-0003-4845-6165
Birgit Engeland  http://orcid.org/0000-0001-9104-7194
Malte Stockebrand  http://orcid.org/0000-0001-9009-137X
Anna Moroni  http://orcid.org/0000-0002-1860-406X
Steven A Siegelbaum  http://orcid.org/0000-0002-0242-7505
Dirk Isbrandt  http://orcid.org/0000-0002-4720-1016
Bina Santoro  http://orcid.org/0000-0002-4277-1992

### Ethics
Mouse colonies were maintained both at the University of Cologne and at Columbia University (New York). For research conducted in Cologne, all experiments were in accordance with European, national and institutional guidelines and approved by the State Office of North Rhine-Westphalia, Department of Nature, Environment and Consumer Protection (LANUV NRW, Germany; reference number 81-02.04.2018.A085). For research conducted in New York, all animal experiments were conducted in accordance with policies of the NIH Guide for the Care and Use of Laboratory Animals and the Institutional Animal Care and Use Committee of Columbia University (IACUC protocols AABL5560, AABL5563, AABI2614 and AAAX6450).

### Decision letter and Author response
Decision letter https://doi.org/10.7554/eLife.70826.sa1
Author response https://doi.org/10.7554/eLife.70826.sa2

---

## Additional files

### Supplementary files
• Transparent reporting form

• Source code 1. MATLAB code to generate *Figure 4A and C*. Source code used to plot typical examples of seizures, including the electrocorticogram (ECoG) trace and corresponding time–frequency spectrogram, using the custom written function *plot_telemSz_andrea*.

• Source data 1. Microarray-based analysis of differential ion channel gene expression in $Hcn1^{GD/+}$ and $Hcn1^{MI/+}$ mice. Average intensity (log2), fold change, p-value, and false discovery rate (FDR)-corrected p-value (considering all 22,206 probe sets) were derived using an Affymetrix-based screening of hippocampal tissue from $Hcn1^{GD/+}$, $Hcn1^{MI/+}$, and WT mice (n = 4 for each group) for 112 candidate genes, representing all main families of voltage-gated ion channels (see 'Materials and methods'). Top 10 hits, ranked by p-value, are shown for each comparison. Values for GFAP and vimentin, two genes with an expected increase in hippocampal expression as a result of reactive gliosis (*Figure 4—figure supplement 1*, *Escartin et al., 2021*; *Stringer, 1996*), are also shown for reference. Note that both markers showed a nearly approximately twofold increase in $Hcn1^{GD/+}$ animals (with FDR-corrected p-value<0.05 in the case of vimentin), while negligible changes were observed in $Hcn1^{MI/+}$ animals.

### Data availability
All data generated or analysed during this study are included in the manuscript and supporting files. Source data files have been provided for Figures 2, 3, 5, 9 and 10. Microarray expression data have been deposited in GEO under accession code GSE209630.

The following dataset was generated:

| Author(s) | Year | Dataset title | Dataset URL | Database and Identifier |
|---|---|---|---|---|
| Merseburg A, Stockebrand M, Isbrandt D, Santoro B | 2022 | Expression data from the hippocampus of epileptic mice carrying two different pathogenic de novo mutations in the hyperpolarization-activated, cyclic nucleotide-gated, non-selective cation channel gene Hcn1 in comparison to wildtype controls | https://www.ncbi.nlm.nih.gov/geo/query/acc.cgi?acc=GSE209630 | NCBI Gene Expression Omnibus, GSE209630 |

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
