## [Editor Report]

This is an important study into the pathogenic role of two distinct sequence alterations in Hcn1 and extends from insights into ion channel physiology all the way to characterizing the animals' spontaneous seizure phenotype. The authors convincingly show that two clinically relevant sequence alterations in Hcn1 have distinct effects on HCN1 channel trafficking that vary with cellular context. They go on to correlate these changes in trafficking and cellular function to differences in seizure behavior and observe that these changes in Hcn1 lead to paradoxical responses to some antiseizure medications, consistent with clinical observations. Intriguingly, although the corresponding genetic changes produce a profound epileptic encephalopathy in human patients, mixed effects on cognition are seen in mice. A limitation of the study is that this difference is not investigated in more detail, but the authors have certainly made headway towards understanding the role of Hcn1 in certain genetic epilepsies.

---

## [Decision Letter]

**Decision letter after peer review:**

Thank you for submitting your article "Seizures, behavioral deficits and adverse drug responses in two new genetic mouse models of HCN1 epileptic encephalopathy" for consideration by *eLife*. Your article has been reviewed by 3 peer reviewers, including Dane Michael Chetkovich as Reviewing Editor and Reviewer #1, and the evaluation has been overseen by John Huguenard as the Senior Editor. The following individual involved in review of your submission has agreed to reveal their identity: Tallie Z Baram (Reviewer #3).

Essential revisions:

1) Please in discussion expound on the limitation to interpretation of data given the lack of physiological evaluation of PV neurons and direct evaluation of HCN channels in dendrites. Similarly please address the possibility that LTG experiments in vitro would not reveal direct effects on the channel in the context of neurons with their native binding partners.

2) Expound on the potential of other conductances that could have been impacted that may explain unexpected intrinsic excitability changes.

3) Please change instances of "Mutant" or "mutation" to "pathological sequence variation" in the context of humans, and revise other anthropomorphized terms referring to animal behavior as reflecting human psychiatric disease.

4) Address claims of differences with requested statistics.

5) Consider approach to consistently co-identify analogous mutations to avoid confusing readers with switches between mouse and human.

6) See reviewer 1 comments regarding figure 6. Figure 6 should be reproduced with PV costaining and uniform high-power magnifications with some attempt at quantitation.

7) Present functional tests of hippocampus or cortex dependent behaviors, regardless of the outcome. At a minimum, please modify the text to eliminate redundant panels in Figure 2. and downplay the cerebellar emphasis. (see Reviewer 3 comments 2a-c for potential discussion points.

*Reviewer #1 (Recommendations for the authors):*

They study the impact of the mutations on the levels and distribution of the channel and identify abnormal targeting of the channels to among other findings, axon terminals of basket cell interneurons.

Seizures were found to be exacerbated by Na^+^ channel blockers lamotrigine and phenytoin, consistent with clinical reports for patients with pathological HCN1 sequence variation, an important feature of good pre-clinical models.

Finally, the report identifies changes to HCN1 antagonist sensitivity in the variant channels.

The authors suggest this justifies screening mutated channels to identify novel compounds with diverse mechanisms of action.

Abstract

The abstract is clear and concise and communicates the major findings of the paper.

One suggested style consideration: "In line with clinical reports from HCN1 patients" Would change to "patients with pathological HCN1 sequence variation." Would consider this in lieu of mutation throughout the article as many physicians and patient advocacy groups take issue with negative connotations of "mutation" and "mutant" in the context of animal work as applied to patients.

Introduction

The authors review of the background and problem is thorough; In Para 4, they address the conundrum that mice might not be able to recapitulate the epilepsy phenotype seen with certain variants. Here would be appropriate to mention that HCN1 KO mice also lack seizures. This suggests it is possible that most of the pathological HCN variants impart gain of function directly on the channel or impacts non-HCN targets (distinct from simple loss of channel function), as well as that gain of function might manifest in seizures distinctly in mouse and man.

Regarding the approach: the choice to generate mice in which phenotypes analogous human variations were replicated in multiple patients and the phenotype was severe was logical and a good example to others about increasing odds of success.

Results

General phenotypic features and gross brain anatomy

Although it looks true, it is not clear statistics were employed to assert death of males significantly more than females; "there was a clear split in Hcn1M142I/+ animals with females more affected than males and most deaths occurring at an age older than three months (Figure 1C, right)." What analysis was done to conclude this?

Page 6 line 1; would not use "increased anxiety" as it anthropomorphizes animal behavior. "considered a potential anxiety-like behavior in mice"

They find complex effect of the two variants on phenotype, with an overall more severe impact on the general fitness of Hcn1G380D/+ compared to Hcn1M142I/+ animals, reflected in smaller body size, reduced brain size, as well as altered gait and locomotion.

Hcn1G380D/+ and Hcn1M142I/+ mice show spontaneous convulsive seizures

The characterization of epilepsy in these mice is solid. The case is made for being a good model of the human disease.

Effects of mutations on HCN1 expression and the intrinsic properties of hippocampal

CA1 pyramidal neurons

Last paragraph of page 7: The switch between mouse and human numbering is accurate throughout, but it is distracting.

This work introduces nicely prior in vitro studies of the two variants, one with predicted "loss of function" impact on physiology and the other with "gain of function. This study, made possible by the novel animals, extends these prior in vitro approaches into the native setting where there will be TRIP8b and other factors:

One concern regarding statistical approach to demonstrate difference between experimental groups: "Labeling revealed a substantial decrease in overall HCN1 protein levels in Hcn1G380D/+ brains, with a smaller but still significant decrease observed in Hcn1M142I/+ brains (Figure 4B). How were the groups compared with each other?

Eliminated Sag in 380: speculate decreased protein, dom neg effect on function, or the basal leak.

Cells were depolarized, c/w basal leak current, this was absent in the M142 cells.

It is noted that input resistance was unaltered in both lines; A caveat should be mentioned that this was in somatic recordings and it is possible input resistance in distal dendrites is not reflected at the soma.

They found impaired ability to fire Action potentials in both lines and some differences is seizure frequency and severity between the lines. "The reduced ability of Hcn1G380D/+ pyramidal neurons to maintain sustained firing may, in fact, act to limit circuit hyperexcitability in these animals – as reflected in the lower average maximum seizure grade reached by Hcn1G380D/+ mutants. The difference was 76% vs 37%. I am unsure what statistics were used to demonstrate that this was a significant difference. Please clarify.

Severe impairment in the axonal localization of HCN1 protein in Hcn1G380D/+ PV+

Interneurons:

The authors indicate the HCN staining is lost from Cerebellar basket cells Pinceau in G380D, but appears normal in M142. In the chosen images (Figure 5) to the eye it appears M124I staining is less extensive than WT, at least for HCN and Kv1.2. This is difficult to quantitate, and electrophysiology would help distinguish the 2 lines from each other and to WT with respect to function.

"The profound disruption of normal pinceau architecture in cerebellar basket cells suggests that, in the case of HCN1-G380D, misfolding of the mutated protein may be causing cellular stress (presumably endoplasmic reticulum stress, due to impaired trafficking and/or increased protein degradation). Such toxicity effects may explain the higher severity of the Hcn1G380D/+ phenotype, which includes altered anatomy and behaviors beyond the occurrence of seizures and epilepsy. This conclusion is speculative and was not addressed experimentally. The use of the word architecture here should be reconsidered as they did not address structure per se; Along these lines it would be important to have another pinceau-expressed marker to demonstrate architecture is actually normal.

Figure 6 is not convincing. It would be good to see high power images (of the same magnification and laser intensity across samples) with PV+ terminals co-stained. Clearly there are HCN-labelled terminals in untransfected G390D mice, and although fainter and at lower magnification in the G380D transfected slices. Overexpression can be variable; whereas the high-power image of G380D shows a bright soma, there are also bright soma stained in the low power WT image. At a minmum this figure should be reproduced with PV costaining and uniform high-power magnifications with some attempt at quantitation.

Paradoxical response of Hcn1G380D/+ and Hcn1M142I/+ mice to antiseizure medications:

These experiments and figures important in demonstrating pharmacosensitivity between human and mouse variants.

The authors did important clarifying work on lamotrigine to show not a direct HCN agonist, which answers an important question in the field.

Reduced efficacy of HCN/Ih-blocking compounds on HCN1-G391D channels and

Hcn1G380D/+ neurons:

The exploration of impact of channel inhibitors on HCN is interesting and allows elegant structure/function discussion, but it is not clearly related to the previously proposed mechanism or central to the paper. If G391D protein is substantially reduced and Ih absent from neurons, even with the leak, it seems counterintuitive to want to block the channel as a therapeutic approach. It would seem a more interesting and informative experiment to also test ZD in the M142 mice, which in vitro exhibit gain of function but in vivo seem underexpressed.

Discussion

The discussion is overall well written and reiterates the conclusions drawn from data.

P16 last para:

"These results underscore how mutations in voltage-gated ion channels can fundamentally alter brain development". It is not clear that brain development was studied; these were functional changes that could impact development.

Last Para of P 18; There is an emphasis on the discernable phenotype differences of the mutants; again, what is the statistical evidence for severity differences?

P19 Para 1; the statement that the different mutations may explain phenotypic variances needs to be tempered with acknowledgement that one often sees GEFS+ or DEE or non-penetrance in the families harboring the same variant (for HCN1 and for pathological variations). The authors own data identified animals harboring the same mutations but with few or no seizures.

P19 last para: Authors focus on gain of function as a target; however, their putative gain of function mouse M142I actually demonstrated loss of function in vivo. So there does not appear to be clear rationale for targeting gain of function as nice as it might be to have a non-pore blocker. Should we not be considering molecules that restore the loss of function, either by targeting the channel itself or trafficking mechanisms?

An important question and logical conclusion from the presented data is left unexplored:

The authors demonstrated that variant HCN leads to loss of KV1.2, ATPase and HCN in terminals. What about Na^+^ channels? Authors note that " Interestingly, the axon-specific expression of HCN1 channels in PV+ interneurons matches the similarly polarized subcellular distribution of voltage-gated Na^+^ channels" Is there a loss of Na^+^ channels in HCN variants that mirrors the SCN1A loss in heterozygous ko mice or Dravet's (and thereby explain the similar human AED sensitivity in the HCN variants?). This seems an important question to answer with these mice.

Overall this is well done.

This reviewer's observations are made in the public review.

A few other style Comments:

"…is directly due to "leaky" HCN1 channels or is a secondary response to changes

in other conductances, further underscoring the need for a radically different approach in dealing both with the therapeutic treatment and experimental investigation of HCN1-

linked DEEs." The second part of that sentence is somewhat hyperbolic and seems to be a nonsequitur to the first. What is the proposed radically different approach? Treatment now is agnostic to HCN channel function and empirically is to avoid using Na^+^ blockers. I think this is trying to say further investigation is needed to understand exact mechanisms, which may be distinct from modulating the current.

Why use the term "developmental trajectories"? This paper did not study development. Better perhaps to use epilepsy phenotypes or network physiology, etc

*Reviewer #2 (Recommendations for the authors):*

The manuscript represents a valuable contribution as is.

However the lack of data on HCN1 channel biophysical parameters in native tissue is a deficit and prevents comparison with a previously published study on a different HCN1 mutant.

The data on lamotrigine and ZD7288 effects on heterologously expressed channels is interesting but a little tangential as without comparison to Ih in native tissue, it is hard to know how to interpret these findings, and there is little argument that these drugs are not highly selective for HCN channels.

*Reviewer #3 (Recommendations for the authors):*

The work is overall terrific and rigorous, and the majority of the conclusions are robust. Yet, there are several issues that require addressing:

a. The authors characterize cerebellum-dependent functional deficits in the mutant mice, basing their studies on the high expression levels of HCN1 in cerebellum, citing Notomi and Shigemoto, They do not present phenotypic deficits in function ascribed to hippocampus or cortex. However:

1. Notomi and Shigemoto state: "Immunoreactivity for HCN1 showed predominantly cortical distribution, being intense in the neocortex, hippocampus, superior colliculus, and cerebellum" ( abstract and Table 1).

2. Importantly, the seizures of the HCN1 mutant mice are unlikely to arise from the cerebellum, and the encephalopathies elements of HCN1-related neonatal epileptic encephalopathies clearly derive from cortex and hippocampus.

Therefore, it should be excellent if the authors presented functional tests of hippocampus or cortex dependent behaviors, regardless of the outcome. At a minimum, they should modify the text, eliminate redundant panels in Figure 2. and downplay the cerebellar emphasis.

b. The authors base their proposed mechanism for the pro-epileptic effects of the mutation on the notion that HCN1 Channels are localized to axons only of PV interneurons. Whereas this fact may be true for the adult, during development, axonal targeting is not unique to basket-type interneurons. It is observed in the developing hippocampal circuit, in medial entorhinal cortex neurons innervating dentate gyrus granule cells, i.e., the perforant path. Have the authors looked at axonal targeting in this region in the mutant mice during appropriate developmental stages? Its absence might modulate the firing of GCs, specifically during development (Bender et al., J Neurosci 2007). At a minimum this point merits discussion.

c. The authors identify developmental epilepsies in the mutants. Therefore, they shouldn't disregard the distinct developmental profiles for HCN1 and the other HCN subunit (e.g.,Surges et al., 2006). First, HCN1 may not function in isolation, and seizures might trigger heteromer formation (e.g., Brewster et al., 2005). In addition, might expression of other subunits 'take over' some functions of HCN1 (even without overexpression). including HCN1, and these profiles might contribute to age-specific defects leading to seizures. This point merits discussion.

d. Whereas the focus of this paper is on the role of genetic mutations in HCN1 in epilepsy, the paper may be enriched by being placed in the context of the overall contributions of HCN1 channels to human epilepsy, including "acquired epilepsy"" via potential epigenetic changes in the expression of normal HCN channels (Bender et al., 2003 and others). They might cite recent reviews of the role of HCN channels in epilepsy (e.g., Brennan et al., 2016).

---

## [Author Response]

Essential revisions:1) Please in discussion expound on the limitation to interpretation of data given the lack of physiological evaluation of PV neurons and direct evaluation of HCN channels in dendrites. Similarly please address the possibility that LTG experiments in vitro would not reveal direct effects on the channel in the context of neurons with their native binding partners.

We have added the required caveats in the Discussion (pages 20-21) and the Results section (pages 10 and 17). Please see responses to Reviewers #1 and #2.

2) Expound on the potential of other conductances that could have been impacted that may explain unexpected intrinsic excitability changes.

Prompted by the reviewer’s comment, we performed additional analyses on our electrophysiology datasets and carried out a new microarray-based screening of bulk hippocampal tissue from our mutant mouse lines to probe for potential secondary changes in ion channel expression (Figure 5 —figure supplement 1 and Source Data 1). We have added new text to emphasize how potential changes in other conductances may impact neurons’ intrinsic properties in our mice (pages 10-11). For a detailed description of the results and modifications introduced, please see the response to Reviewer #1.

3) Please change instances of "Mutant" or "mutation" to "pathological sequence variation" in the context of humans, and revise other anthropomorphized terms referring to animal behavior as reflecting human psychiatric disease.

We thank the Editors for these important recommendations, and we have modified the manuscript text throughout to improve language and terminology.

4) Address claims of differences with requested statistics.

Statistical analysis was added for all datasets and questions highlighted by Reviewer #1 (please see the response to Reviewer #1 for a detailed description of changes).

5) Consider approach to consistently co-identify analogous mutations to avoid confusing readers with switches between mouse and human.

We have eliminated positional numbering information whenever referring to the mouse variants. We maintained, however, the appropriate nomenclature for the relevant human variants (e.g., p.G391D and p.M153I) given that new *HCN1* variants are being identified every year, and we want to retain clarity about what specific human variants represent the focus of our current paper.

6) See reviewer 1 comments regarding figure 6. Figure 6 should be reproduced with PV costaining and uniform high-power magnifications with some attempt at quantitation.

New experiments were performed, and both requests have been fulfilled (see new Figure 7 and Figure 7 —figure supplements 1 and 2). The new PV co-staining data and fluorescence intensity quantification of virus-driven HCN1 expression confirm and expand our previous results, and we are grateful to the reviewer for this recommendation. Please see the response to Reviewer #1 for a detailed description of the results and modifications introduced.

7) Present functional tests of hippocampus or cortex dependent behaviors, regardless of the outcome. At a minimum, please modify the text to eliminate redundant panels in Figure 2. and downplay the cerebellar emphasis. (see Reviewer 3 comments 2a-c for potential discussion points.

We performed new behavioral experiments, including an assessment of forebrain cortex-dependent function, and have added these in modified Figure 2 and new Figure 3. These experiments indeed revealed specific impairments in the performance of *Hcn1^GD/+^* but not *Hcn1^MI/+^* animals. The manuscript text has been correspondingly revised (see Results, pages 6 and 7) with added discussion, as requested. A description of the modifications introduced is also provided in the responses to Reviewers #2 and #3 below.

Reviewer #1 (Recommendations for the authors):AbstractThe abstract is clear and concise and communicates the major findings of the paper.One suggested style consideration: "In line with clinical reports from HCN1 patients" Would change to "patients with pathological HCN1 sequence variation." Would consider this in lieu of mutation throughout the article as many physicians and patient advocacy groups take issue with negative connotations of "mutation" and "mutant" in the context of animal work as applied to patients.

We are grateful for this suggestion and have modified the text throughout. We now use the term “pathogenic sequence variation” in all instances, in line with human genetics terminology.

IntroductionThe authors review of the background and problem is thorough; In Para 4, they address the conundrum that mice might not be able to recapitulate the epilepsy phenotype seen with certain variants. Here would be appropriate to mention that HCN1 KO mice also lack seizures.

We thank the reviewer for the suggestion and now discuss that HCN1 KO mice do not show spontaneous seizures in the Introduction (page 4).

ResultsGeneral phenotypic features and gross brain anatomyAlthough it looks true, it is not clear statistics were employed to assert death of males significantly more than females; "there was a clear split in Hcn1M142I/+ animals with females more affected than males and most deaths occurring at an age older than three months (Figure 1C, right)." What analysis was done to conclude this?

We have now added a statistical analysis using Fisher’s exact probability test, which confirms our conclusion regarding differential death rates of male vs. female *Hcn1^MI/+^* but not *Hcn1^GD/+^* animals (page 5).

Page 6 line 1; would not use "increased anxiety" as it anthropomorphizes animal behavior. "considered a potential anxiety-like behavior in mice"

We agree that this is a more accurate terminology and have modified the text accordingly.

Last paragraph of page 7: The switch between mouse and human numbering is accurate throughout, but it is distracting.

We have adjusted any references to the mouse variants by omitting the numbering for increased simplicity (see new text on page 5).

One concern regarding statistical approach to demonstrate difference between experimental groups: "Labeling revealed a substantial decrease in overall HCN1 protein levels in Hcn1G380D/+ brains, with a smaller but still significant decrease observed in Hcn1M142I/+ brains (Figure 4B). How were the groups compared with each other?

We have added a new analysis to address this question. We normalized the fluorescence signal in *Hcn1^GD/+^* and *Hcn1^MI/+^* to the WT signal in each cohort and compared the change in fluorescence intensities between the two lines as a fraction of corresponding control intensities. Statistical analysis using 2-way repeated measures-ANOVA corroborated our interpretation that the reduction in HCN1 protein fluorescence intensity is indeed stronger in *Hcn1^GD/+^* as compared to *Hcn1^MI/+^* pyramidal neurons (new panel in Figure 5B, formerly Figure 4B).

It is noted that input resistance was unaltered in both lines; A caveat should be mentioned that this was in somatic recordings and it is possible input resistance in distal dendrites is not reflected at the soma.

The Reviewer raises a valid point, and we have introduced the appropriate caveat (page 10).

They found impaired ability to fire Action potentials in both lines and some differences is seizure frequency and severity between the lines. "The reduced ability of Hcn1G380D/+ pyramidal neurons to maintain sustained firing may, in fact, act to limit circuit hyperexcitability in these animals – as reflected in the lower average maximum seizure grade reached by Hcn1G380D/+ mutants. The difference was 76% vs 37%. I am unsure what statistics were used to demonstrate that this was a significant difference. Please clarify.

We have added a statistical analysis demonstrating that the relative difference of 76% vs. 37% of cells showing abnormal AP firing patterns in *Hcn1^GD/+^* compared to *Hcn1^MI/+^* animals is significant (P=0.006, Fisher’s exact probability test, see page 11).

Severe impairment in the axonal localization of HCN1 protein in Hcn1G380D/+ PV+Interneurons:The authors indicate the HCN staining is lost from Cerebellar basket cells Pinceau in G380D, but appears normal in M142. In the chosen images (Figure 5) to the eye it appears M124I staining is less extensive than WT, at least for HCN and Kv1.2. This is difficult to quantitate, and electrophysiology would help distinguish the 2 lines from each other and to WT with respect to function."The profound disruption of normal pinceau architecture in cerebellar basket cells suggests that, in the case of HCN1-G380D, misfolding of the mutated protein may be causing cellular stress (presumably endoplasmic reticulum stress, due to impaired trafficking and/or increased protein degradation). Such toxicity effects may explain the higher severity of the Hcn1G380D/+ phenotype, which includes altered anatomy and behaviors beyond the occurrence of seizures and epilepsy. This conclusion is speculative and was not addressed experimentally. The use of the word architecture here should be reconsidered as they did not address structure per se; Along these lines it would be important to have another pinceau-expressed marker to demonstrate architecture is actually normal.

The Reviewer raises an important point, and we have deleted this sentence from the Results section. We have also added text to clarify that our immunolabeling experiments cannot distinguish between loss of marker expression and loss of axon structure (page 14). Unfortunately, no matter what marker we use, at present, our light microscopy approach will not resolve the question of axonal expression vs. architecture. This distinction will have to await future experiments employing electron microscopy or other suitable techniques.

Reduced efficacy of HCN/Ih-blocking compounds on HCN1-G391D channels andHcn1G380D/+ neurons:The exploration of impact of channel inhibitors on HCN is interesting and allows elegant structure/function discussion, but it is not clearly related to the previously proposed mechanism or central to the paper. If G391D protein is substantially reduced and Ih absent from neurons, even with the leak, it seems counterintuitive to want to block the channel as a therapeutic approach. It would seem a more interesting and informative experiment to also test ZD in the M142 mice, which in vitro exhibit gain of function but in vivo seem underexpressed.

We have added text to clarify the motivation for this experimental dataset (pages 17 and 18), aimed at testing the efficacy of currently available pore blocking compounds on HCN1 variants with a “leaky” conductance. Our paper indeed argues in favor of the hypothesis that cellular stress may be a major component in the pathophysiology linked to the p.G391D variant, reflected partly by the overall decrease in HCN1 protein expression. Since reduced expression of HCN1 protein *per se* does not appear to cause epilepsy, as HCN1 null mice do not have spontaneous seizures, these premises suggest that an antisense oligonucleotide (ASO) approach may be ideal in the case of *Hcn1^GD/+^* mice as it would eliminate the production of misfolded protein. However, our results do not rule out a damaging impact of the “leak” component due to altered HCN current expression. This component has been shown to directly lead to hyperexcitability in *Hcn1^M294L^* mice (Bleakley et al., 2021). Since several of the *HCN1* de novo sequence variants associated with severe clinical phenotypes result in “leaky” HCN channels (Porro et al., 2021), we were curious to establish whether pore blockers would be still effective in this category of variants as proof of principle. As Reviewer #3 pointed out, this information seems to provide valuable information in the larger context of *HCN1*-linked epilepsy treatment options. We hope that the added text will provide further clarification. We did not test ZD7288 on *Hcn1^MI/+^* mice because, in the current clamp recordings, we did not observe significant changes in RMP or input resistance, and also this variant is not expected to affect pore structure.

DiscussionThe discussion is overall well written and reiterates the conclusions drawn from data.P16 last para:"These results underscore how mutations in voltage-gated ion channels can fundamentally alter brain development". It is not clear that brain development was studied; these were functional changes that could impact development.

The point is well taken, and we have eliminated any direct reference to brain development in the *Hcn1^GD/+^* and *Hcn1^MI/+^* mice. As suggested by Reviewer #3, we have added a more general reference to the role of HCN channels in cortical network development (page 19).

Last Para of P 18; There is an emphasis on the discernable phenotype differences of the mutants; again, what is the statistical evidence for severity differences?

Two separate lines of observation lead us to conclude that the phenotype of *Hcn1^GD/+^* mice is more severe than *Hcn1^MI/+^* mice. First, *Hcn1^GD/+^* mice showed a greater number of alterations than WT mice in the parameters examined in Figures 1-3 compared to *Hcn1^MI/+^* mice. Thus, brain size, gait analysis, pole climbing, and spontaneous alternation were all normal in *Hcn1^MI/+^* mice but were altered significantly in *Hcn1^GD/+^* mice. Second, we observed larger changes in HCN1 protein levels and a greater frequency of neurons with abnormal AP firing in *Hcn1^GD/+^* mice compared to *Hcn1^MI/+^* mice. We have added direct comparisons and statistical analysis to illustrate this point (new panel in Figure 5B, and page 11).

P19 Para 1; the statement that the different mutations may explain phenotypic variances needs to be tempered with acknowledgement that one often sees GEFS+ or DEE or non-penetrance in the families harboring the same variant (for HCN1 and for pathological variations). The authors own data identified animals harboring the same mutations but with few or no seizures.

We agree with the Reviewer that there is variability in the frequency of seizures among animals within the same line (Figure 4). However, when it comes to the more severe de novo HCN1 sequence variants (including p.G391D and p.M153I), the clinical phenotype of patients presenting with the same variant is remarkably consistent (Marini et al., 2018; see also page 14). At the same time, it is indeed the case that for less severe HCN1 variants, including familial variants, there is a much wider range and variability in clinical presentation. We have now added text to clarify these distinctions (pages 21-22), in line with the Reviewer’s comments.

P19 last para: Authors focus on gain of function as a target; however, their putative gain of function mouse M142I actually demonstrated loss of function in vivo. So there does not appear to be clear rationale for targeting gain of function as nice as it might be to have a non-pore blocker. Should we not be considering molecules that restore the loss of function, either by targeting the channel itself or trafficking mechanisms?

The Reviewer is correct in pointing out that there is, generally speaking, a “loss” of function in that the sag is reduced and HCN1 protein expression is diminished in both our mutant mouse lines. However, the *Hcn1^GD/+^* line and the *Hcn1^M294L^* line characterized by Bleakley et al., (2021) also show what has been deemed a “gain of aberrant” function, as the current generated by these mutated channels has a prominent voltage-independent component. While pharmacological agents able to improve protein trafficking and reduce cellular stress would certainly seem beneficial, there are certain risks associated with increasing the activity of this inward Na^+^ leak.

An important question and logical conclusion from the presented data is left unexplored:The authors demonstrated that variant HCN leads to loss of KV1.2, ATPase and HCN in terminals. What about Na^+^ channels? Authors note that " Interestingly, the axon-specific expression of HCN1 channels in PV+ interneurons matches the similarly polarized subcellular distribution of voltage-gated Na^+^ channels" Is there a loss of Na^+^ channels in HCN variants that mirrors the SCN1A loss in heterozygous ko mice or Dravet's (and thereby explain the similar human AED sensitivity in the HCN variants?). This seems an important question to answer with these mice.

This is certainly a possibility, and indeed one that we very much favor. We have added a new analysis of AP morphology (Figure 5 – figure supplement 1) and performed a microarray-based experiment to screen for changes in Na+ channel expression (Source Data 1). While these experiments yielded negative results, they do not definitively rule out potential cell-type specific alterations in the function of Na+ channels or other conductances. A more thorough experimental examination of this important question will have to await future studies. We have added text to underscore how changes in other conductances may indeed impact neurons’ intrinsic properties in our mice (pages 10-11).

A few other style Comments:"…is directly due to "leaky" HCN1 channels or is a secondary response to changesin other conductances, further underscoring the need for a radically different approach indealing both with the therapeutic treatment and experimental investigation of HCN1-linked DEEs." The second part of that sentence is somewhat hyperbolic and seems to be a nonsequitur to the first. What is the proposed radically different approach? Treatment now is agnostic to HCN channel function and empirically is to avoid using Na^+^ blockers. I think this is trying to say further investigation is needed to understand exact mechanisms, which may be distinct from modulating the current.

We thank the reviewer for the helpful comment. We have deleted this sentence and offered additional clarification on the intent of our analysis.

Why use the term "developmental trajectories"? This paper did not study development. Better perhaps to use epilepsy phenotypes or network physiology, etc

The point is well taken, and we have taken out any reference to brain development in our mouse lines, as we do not have specific information on the matter. As per Reviewer’s #3 request, we now only mention select relevant literature, to provide some general information regarding the potential role of HCN1 channels during cortical development (page 19).

Reviewer #2 (Recommendations for the authors):The manuscript represents a valuable contribution as is.However the lack of data on HCN1 channel biophysical parameters in native tissue is a deficit and prevents comparison with a previously published study on a different HCN1 mutant.

We agree that obtaining more detailed information about the biophysical properties of the HCN1 variants in native tissue would be of interest. However, technical limitations are known to prevent accurate voltage clamp recordings of hyperpolarization-activated currents from neurons in brain slices, even using cell-attached patch recordings (Williams SR, Wozny C. Errors in the measurement of voltage-activated ion channels in cell-attached patch-clamp recordings. Nat Commun. 2011; 2:242). We have added text in the manuscript to underscore the limitations of our analysis (see also point b above).

The data on lamotrigine and ZD7288 effects on heterologously expressed channels is interesting but a little tangential as without comparison to Ih in native tissue, it is hard to know how to interpret these findings, and there is little argument that these drugs are not highly selective for HCN channels.

We agree with the Reviewer and have added the needed caveat in the appropriate Results section (page 17).

Reviewer #3 (Recommendations for the authors):The work is overall terrific and rigorous, and the majority of the conclusions are robust. Yet, there are several issues that require addressing:a. The authors characterize cerebellum-dependent functional deficits in the mutant mice, basing their studies on the high expression levels of HCN1 in cerebellum, citing Notomi and Shigemoto, They do not present phenotypic deficits in function ascribed to hippocampus or cortex. However:1. Notomi and Shigemoto state: "Immunoreactivity for HCN1 showed predominantly cortical distribution, being intense in the neocortex, hippocampus, superior colliculus, and cerebellum" ( abstract and Table 1).2. Importantly, the seizures of the HCN1 mutant mice are unlikely to arise from the cerebellum, and the encephalopathies elements of HCN1-related neonatal epileptic encephalopathies clearly derive from cortex and hippocampus.Therefore, it should be excellent if the authors presented functional tests of hippocampus or cortex dependent behaviors, regardless of the outcome. At a minimum, they should modify the text, eliminate redundant panels in Figure 2. and downplay the cerebellar emphasis.

Following the Reviewer’s helpful recommendations, we have added new behavioral experiments testing short-term and long-term memory (see new Figure 3) and modified the panels in Fig 2. The manuscript text has been revised accordingly (pages 6 and 7).

b. The authors base their proposed mechanism for the pro-epileptic effects of the mutation on the notion that HCN1 Channels are localized to axons only of PV interneurons. Whereas this fact may be true for the adult, during development, axonal targeting is not unique to basket-type interneurons. It is observed in the developing hippocampal circuit, in medial entorhinal cortex neurons innervating dentate gyrus granule cells, i.e., the perforant path. Have the authors looked at axonal targeting in this region in the mutant mice during appropriate developmental stages? Its absence might modulate the firing of GCs, specifically during development (Bender et al., J Neurosci 2007). At a minimum this point merits discussion.

The Reviewer correctly points out that HCN1 channels are present not only in the axons of PV+ interneurons but also in the axons of certain subclasses of excitatory neurons (see Huang et al., 2011, 2012, and 2019). Regarding axons from medial entorhinal cortex neurons innervating dentate gyrus granule cells, i.e., the perforant path, there is an interesting difference between mice and rats. While HCN1 channel subunits at this site are downregulated in adult rats, they persist in adult mice. This can be seen in the immunostainings shown in Figure 5A (formerly 4A) of the manuscript. Similar to hippocampal PV+ axons in CA3 (Figure 7A, formerly 6A), it can be noted that HCN1 expression in the perforant path is considerably decreased in Hcn1GD/+ mice compared to wildtype and Hcn1MI/+ mice.

c. The authors identify developmental epilepsies in the mutants. Therefore, they shouldn't disregard the distinct developmental profiles for HCN1 and the other HCN subunit (e.g.,Surges et al., 2006). First, HCN1 may not function in isolation, and seizures might trigger heteromer formation (e.g., Brewster et al., 2005). In addition, might expression of other subunits 'take over' some functions of HCN1 (even without overexpression). including HCN1, and these profiles might contribute to age-specific defects leading to seizures. This point merits discussion.

We thank the reviewer for raising this important point and have added text underscoring the potential contribution of altered HCN1 channel function to brain development (page 19) to address this issue, and in accord with the comments raised by Reviewer #1 above.

d. Whereas the focus of this paper is on the role of genetic mutations in HCN1 in epilepsy, the paper may be enriched by being placed in the context of the overall contributions of HCN1 channels to human epilepsy, including "acquired epilepsy"" via potential epigenetic changes in the expression of normal HCN channels (Bender et al., 2003 and others). They might cite recent reviews of the role of HCN channels in epilepsy (e.g., Brennan et al., 2016).

We agree with the Reviewer and now refer to these datasets in the Introduction, citing the excellent review by Brennan et al., 2016 (page 4).